# Inter-comparison of ABL height estimates from different profiling sensors and models in the framework of HyMeX-SOP1

Donato Summa[1,2], Fabio Madonna[1] , Noemi Franco[2], Benedetto De Rosa[1,2] and Paolo Di Girolamo[2*]

[1]Consiglio Consiglio Nazionale delle Ricerche—Istituto di Metodologie per l'Analisi Ambientale
(CNR-IMAA), 85050, Tito Scalo, Potenza,  Italy
[2]Scuola di Ingegneria, Università degli Studi della Basilicata, via Ateneo Lucano 10, 85100 Potenza, Italy

*Correspondence to: Paolo Di Girolamo (paolo.digirolamo@unibas.it)

**Abstract.** This paper reports results from an inter-comparison effort involving different sensors and models used to measure the Atmospheric Boundary Layer height (ABLH). The effort took place in the framework of the first Special Observing Period of the Hydrological cycle of the Mediterranean Experiment (HyMeX-SOP1), with the Raman lidar system BASIL deployed in Candillargues (Southern France) and operating in almost continuous mode over the time period September-November 2012. ABLH estimates were obtained based on the application of the Richardson number technique to Raman lidar and radiosonde measurements and to ECMWF-ERA5 reanalysis data. In the effort we considered radiosondes launched in the proximity of the lidar site, as well as radiosondes launched from the closest radiosonde station included in the Integrated Global Radiosonde archive (IGRA). The inter-comparison effort also includes ABLH measurements from the wind profiler, which rely on the turbulence method, as well as measurements obtained from elastic backscatter lidar signals. The Richardson number approach applied to the on-site radiosonde data is taken as reference. Measurements were carried out throughout the month of October 2012. The inter-comparison is extended to both daytime and night-time data. Results reveal a very good agreement between the different approaches, with values of the correlation coefficient $R^2$ for all compared sensor/model pairs in the range 0.94 to 0.98. Values of the slope of the fitting line in the regression analysis applied to the different sensor/model pairs are in the range 0.91-1.08 for daytime comparisons and in the range 0.95-1.03 for night-time comparisons, which testifies the very small biases affecting all five ABLH estimates with respect to the reference AHBL estimate, with slightly smaller bias values found at night. Results also confirm that the combined application of different methods to distinct sensors and model data allow getting accurate and cross-validated estimates of the ABL height in a variety of weather conditions. Correlations between the ABLH measurements and other atmospheric dynamic and thermodynamic variables, such as CAPE, friction velocity and relative humidity, are also evaluated to infer possible mutual dependences.

## 1. Introduction

The ABL is lowest portion of the atmosphere, directly in contact and influenced by the Earth's surface, which  reacts to the combined action of mechanical and thermal forcing factors. In this layer, turbulent air motion and vertical mixing induced by

shear and buoyancy forces (Stull, 1988) cause rapid fluctuations of physical quantities such as flow velocity, temperature and moisture. The variability of these quantities is frequently considered to estimate the ABLH.

The evolution of the ABL structure and height has an important impact on meteorology. Accurate measurements of the ABLH allow validating air quality and forecast models and improving specific physical schemes, among others, the boundary layer turbulence and shallow convection parameterizations. However, in many cases, the complexity of the phenomena occurring within the ABL and the influence of advection and local accumulation processes prevents an unambiguous determination of the ABLH from e.g. elastic lidar signals, especially when aerosol stratifications are present within the ABL (Haeffelin et al., 2012). Various methods have been reported in the literature to estimate the ABLH from the vertical profiles of different atmospheric variables. The five most used approaches are: 1) the Turbulence method (Stull, 1988), 2) the Temperature gradient method (Bianco and Wilczak, 2002, Zeng et al. 2004, Seidel at al. 2012), 3) the Richardson number method (Joffre et al., 2001, Sicard et al., 2006), 4) the wind (Melgarejo and Deardorff,1974) and wind shear (Hyun et al., 2005) profile methods, and 5) the combined cloud top and relative humidity method (Lenschow et al., 2000), this latter being applicable only in the presence of stratocumulus-topped boundary layers. A review of these different methodologies was reported by Dai et al. (2014a). In the present research effort we compare ABLH measurements obtained through the application of the Richardson number technique to a variety of sensors and model data, namely the Raman lidar BASIL, radiosondes and ECMWF-ERA5 analysis data. These results are also compared with ABLH measurements from the wind profiler and from elastic backscatter lidar signals.

The Richardson number method relies on the identification of Richardson number gradients, which is an important diagnostic indicator of dynamic flow stability. This method assumes the ABLH to be the level where the so called "bulk Richardson number" exceeds a specific threshold value, $Ri_{bc}$. The vertical profile of $Ri_b$ can be calculated from wind speed and potential virtual temperature profiles, as originally reported in Hanna (1969) and extensively described in e.g., Stull (1988) and Garratt (1994). As the estimate of the Richardson number requires information on both wind and thermodynamic profiles, the application of this approach to the Raman lidar relies on thermodynamic profile measurements from this sensor and wind measurements from the simultaneous and co-located radiosondes, as the wind measurements are not available from the Raman lidar.

ABLH measurements carried out by the wind profiler rely on the turbulence method, which identifies the ABLH as the depth of the lowest continuous turbulence layer (Stull, 1988). The turbulent region is determined by tracking the fluctuations of the different wind components ($U$, $V$, and $W$), for example through a high-pass wavelet filter (Wang et al. 1999). Such fluctuations can be identified in wind lidar and wind profiler data, but measurements can also be performed with radiosondes and tethered balloons or in-situ sensors on-board scientific aircrafts.

An alternative and effective method to determine the ABLH rely on the strong sensitivity of elastic backscatter lidar signals to suspended aerosol particles and their gradient and on the circumstance that aerosols, being more abundant within the ABL than in the free troposphere, can act as tracers of atmospheric motions. Such approach, extensively used in the present inter-comparison effort, both in day and night time, is described in detail in the following section.

Another convenient, reliable and widely used approach to determine the ABL height and structure both in daytime and night time is based on the identification of local maxima in potential temperature vertical gradient profiles as measured by meteorological radiosondes (Cramer, 1972; Oke, 1988; Stull, 1988; Sorbjan, 1989; Garratt, 1992; Van Pul et al., 1994; De Wekker et al., 1997; Martucci et al.,2007; Behrendt et al., 2011).

This paper does not represent the first research effort dedicated to an extensive inter-comparison of different ABLH sensors/methodologies. In a previous paper, Seibert et al. (2000) compared different methods to estimate the ABLH from radiosounding data and other instruments and carefully examined advantages and shortcomings of the investigated approaches. In the present paper ABLH estimates obtained through the application of the Richardson number technique to two sensors (Raman lidar and radiosondes) and to the ECMWF-ERA5 model reanalysis data are compared with ABLH

measurements from the wind profiler and from elastic backscatter lidar signals. The capability of the Raman lidar BASIL to perform high-resolution and accurate profile measurements of atmospheric temperature and water vapour, as well as particle backscatter profile measurements, allows to obtain ABLH estimates based on both the application of the Richardson number approach and the use of elastic backscatter lidar signals.

      However, accurate estimates of the ABL height and structure in complex terrains or under complex meteorological

conditions remain problematic. A variety of authors have tried to address this challenging issue. Among others, Herrera-Mejía and  Hoyos (2019) studied the  spatio-temporal evolution of the ABL in a narrow, highly complex terrain located in the Colombian Andes, where convective activity, as a result of aerosol dispersion, increases the uncertainty affecting the estimate of ABLH. Staudt (2006) provided a comprehensive analysis of the ABLH variability over complex terrains in the Bavarian Alpine foreland, based on the use of multi-sensor data collected during the field experiment SALSA 2005. Che and

Zhao (2021) assessed the effectiveness of different approaches to characterize the summer ABLH variability  over the Tibetan Plateau, this region being characterized by elevations exceeding 4000 m and complex land surface processes and boundary layer structures.

      Coming to the characterization of the ABLH in cloudy conditions, Dang et al. (2019) investigated different approaches, with a specific focus on reducing the interference of the residual and cloud layers on ABLH determination. Manninen et al.

(2019) demonstrated the capability of Doppler lidars to determine the ABLH in both clear-sky and cloud-topped conditions, with some reservations in precipitation. Furthermore, Liu et al. (2022) proposed an approach to estimate ABLH from elastic backscatter lidar data under complex atmospheric conditions based on the use of machine learning methods.

      However, both the results from these previous papers and the conclusions reached in the present paper clearly indicate that a proper characterization of the ABL height and structure in all weather conditions requires the combined application of

different approaches and data sets. This approach allows to overcome the possible dependence of each single sensor/method from a specific meteorological parameter, thus drastically reducing potential biases affecting ABLH estimates (Dai et al., 2014b). Additionally, multi-sensor approaches have demonstrated to be more robust and better performing in variable stable and unstable weather conditions (Joffre et al. 2001).

The paper outline is the following. Section 2 illustrates the different methods considered in the present research effort to determine the ABLH. Section 3 shortly describes the profiling sensors and model data involved in the inter-comparison effort. Section 4 illustrates the results from the inter-comparison effort. Finally, section 5 provides a summary, concluding remarks and indications for possible future follow-on studies.

## 2 Methods considered for the determination of the ABLH

### 2.1 Richardson number method

This method assumes the ABLH to be the level where the so called "bulk Richardson number" exceeds a specific threshold value, $Ri_{bc}$. $Ri_b$ at height $z$ can be calculated from the wind speed and the potential virtual temperature values at $z$ and at surface level, as originally reported in Hanna (1969) and extensively described in e.g., Stull (1988) and Garratt (1994). In the present research effort the bulk Richardson number has been computed through the following expression:

$$Ri_b(z) = \frac{\left(\frac{g}{\theta_v(0)}\right)\left(\theta_v(z) - \theta_v(0)\right) z}{u^2(z) \quad + v^2(z)} \tag{1}$$

where $\theta_v(0)$ and $\theta_v(z)$ is the virtual potential temperature at surface and at height $z$, respectively, $\frac{g}{\theta_v(0)}$ is the buoyancy parameter, and $u^2(z)_z$ and $v^2(z)$ are the horizontal wind-speed components at height z, respectively.

The threshold Richardson number $Ri_{bc}$ have been reported in a variety of literature papers (Zilitinkevich and Baklanov, 2002; Jericevic and Grisogono, 2006; Esau and Zilitinkevich, 2010). Reported values are in the range 0.15 to 1.0, with most widely used values in the range 0.25-0.5. One important cause of the large variability of $Ri_b$ is the thermal stratification in the ABL. For example, Vogelezang and Holtslag (1996) reported the $Ri_{bc}$ values of 0.16–0.22 in a nocturnal strongly stable ABL and 0.23–0.32 in a weakly stable ABL. For unstable ABLs, a $Ri_{bc}$ value larger than 0.25 is usually needed (Zhang et al., 2014). The ABLH is found by assessing the altitude level where $Ri_b(z)$ reaches the $Ri_{bc}$. In the present research effort we are considering $Ri_{bc}$=0.25 in stable boundary layers and $Ri_{bc}$=0.45 in unstable boundary layers.

In the present research effort we generally considered the layers to be stable during the evening, night and early morning periods, while layers were assumed to be unstable during the late mornings and afternoons in clear air conditions. The period September-November 2012 was selected for the field campaign because of the specific interest and focus on convection of the Hydrological cycle of the Mediterranean Experiment (HyMeX) first Special Observing Period (SOP1). Within this three-month period, Intense Observation Periods (IOPs) were typically selected when convective instability conditions were present. Thus, the number of cases characterized by unstable layers is predominant. Conditions are found to be persistently unstable during the IOP on October 18-19 and in the period October 16-22. The presence of stable or unstable conditions was verified based on the use of a specific atmospheric stability index, i.e. the "lifted index", obtained from ERA5 model analysis data (see forthcoming figure 5).

The main uncertainty affecting the ALBH estimate based on the application of Richardson number is related to its sensitivity to atmospheric stability conditions and to sounding vertical resolution (Seidel et al., 2012). The ABLH is obtained from both

the radiosonde and model data using the above described algorithm, which is applied from the surface upwards. In case the ABLH falls in between two levels, a linear interpolation is applied to determine its exact position.

In the determination of the bulk Richardson number through the expression (1) the virtual potential temperature and horizontal wind-speed components profiles are needed. The vertical profile of the potential virtual temperature can be expressed as:

$$\theta_v(z) = T(z) \left(\frac{P_0}{P(z)}\right)^{0.286} [1 + 0.622 * \chi_{H2O}(z)] \tag{2}$$

where $P_0$ is the standard pressure (1 atm), $T(z)$ is temperature profile, $P(z)$ is the pressure profile and $\chi_{H2O}(z)$ is the water vapour mixing ratio profile.

Vertical profiles of $T(z)$, and $\chi_{H2O}(z)$ are available from all sensors/models involved in the inter-comparison effort, namely the Raman lidar, the radiosondes launched on-site and those and from the closest IGRA radiosonde station and the ECMWF-ERA5 analysis data, with the only exception of the wind profiler. For most sensors, vertical profiles of $P(z)$ are obtained by vertically extrapolating the surface pressure value $P_0$. The vertical profiles of the horizontal wind-speed components $u(z)$ and $v(z)$, which are needed to quantify the bulk Richardson number are also available from the same sensors/models, with the only exception of the Raman lidar, which only provides the humidity and temperature profile measurements needed to determine the virtual potential temperature profile. In this case, the computation of the Richardson number is completed with the inclusion of wind-speed profile measurements from the simultaneous and co-located radiosondes launched in Candillargues.

## 2.2 Determination of the ABLH from elastic backscatter lidar measurements

Lidar systems are presently able to provide continuous measurements of atmospheric variables, such as particle backscatter, temperature or water vapour concentration profiles, and thus allow providing continuous measurements of the ABLH. In the present research effort we consider Raman lidar measurement from BASIL collected in Southern France in the period September-November 2012 in the frame of HyMeX-SOP1. We focus our attention on the measurements carried out during October 2012.

There are several methodologies to determine the ABLH from elastic lidar signals, which rely on the circumstance that aerosols are more abundant within the ABL than in the free troposphere and they can act as tracers of atmospheric motions. An extensively used methodology relies on the detection of vertical gradients in elastic backscatter lidar signals, associated with aerosol concentration gradients.

The elastic lidar equation, expressed in terms of number of collected photons as function of height, is defined as:

$$P_{\lambda_0}(z) = P_{\lambda_0} O(z) \frac{A}{z^2} \left[\beta_{par}(z) + \beta_{mol}(z)\right] T_{mol}^2(z) \, T_{par}^2(z) + P_{bgd} \tag{3}$$

where $\lambda_0$ is the emitted and received lidar wavelength, respectively, $z$ is the vertical height, $P_{\lambda_0}$ is the number of laser emitted photons, $O(z)$ is the overlap function, $A$ is the telescope collection area, $\beta_{mol}(z)$ and $\beta_{par}(z)$ are the backscatter coefficient for

molecules and particles, respectively, $T_{mol}(z)$ and $T_{par}(z)$ represent the molecular and particle contribution to atmospheric transmissivity, respectively, and $P_{bgd}$ is the background signal associated with solar irradiance and detectors' noise. In order to compress the signal dynamical variability and define an uncalibrated quantity proportional to total (molecular + particle) attenuated backscattering coefficient, it is often preferable to make use of the range-corrected signals (RCSs), which is defined as:

$$RCS(z) = \left[ P_{\lambda_0}(z) - P_{bgd}(z) \right] z^2 \tag{4}$$

As the larger vertical variability of the RCS is associated with aerosol vertical gradients, the ABLH is estimated from the height derivative of RCS through the expression:

$$ABLH = min\left\{ \frac{d}{dz} \left[ \log(RCS(z)) \right] \right\} \tag{5}$$

Transitions between different aerosol layers are identified with the minima in expression (5), with the highest amplitude minimum, i.e. the largest aerosol gradient, typically indicating the ABLH. This approach relies on the strong sensitivity of elastic backscatter lidars to suspended aerosol particles and their gradient and on the capability of aerosols to act as tracers of atmospheric motions. It is to be specified that ABLH estimates determined through this approach identify the convective boundary layer height during the day, when convective activity is on, and the residual layer during the early morning, the late afternoon and the night, when convective activity is strongly reduced or suppressed.

For the specific purposes of our present study, the elastic backscatter signal at 355 nm, $P_{355}(z)$, is considered in expression (3) (Summa at al., 2013; Vivone et al., 2021). Overlap effects affect lidar signals in the lower few hundred meters, but have marginal effects on gradient measurements and consequently the accuracy of ABLH estimates. In fact, overlap effects may determine a compression of the elastic signal dynamics, and consequently a reduction of the amplitude of the detected range-corrected signal gradients, but will not cancel these gradients, making the detection of the ABLH still effective. Overlap effects are more pronounced when determining the night-time ABLH, especially when this height is within a few tens of meters and may fall in the lidar blind region.

## 3 Profiling sensors and model data involved in the inter-comparison effort

This section illustrates the characteristics, specifications and performance of the ground-based instruments and models used in the present research effort to estimate the ABLH: a Raman Lidar system, a UHF wind profiler, co-located radiosoundings and those from the nearest site available in the IGRA. Instrumental pros and cons and potential sources of bias of the different instruments/reanalysis data are reported in Table 1.

### 3.1 BASIL System

The University of BASILicata ground-based Raman Lidar system (BASIL) was deployed in the Cévennes-Vivarais (CV) site (Candillargues, Southern France, Lat: 43°37' N, Long: 4° 4' E, Elev: 1 m, figure 1) and operated between 5 September and 5 November 2012, collecting more than 600 hours of measurements, distributed over 51 measurement days and 19

intensive observation periods (IOPs, Di Girolamo *et al.*, 2016; Stelitano *et al.*, 2019). BASIL is capable to perform high-resolution and accurate measurements of atmospheric temperature and water vapour, both in daytime and night-time, based on the application of the rotational and vibrational Raman lidar techniques, respectively, in the UV (Di Girolamo *et al.*, 2004, 2009, 2017). This measurement capability makes BASIL an effective tool for the characterization of water vapour inflows in Southern France, which are a key ingredient of heavy precipitation events taking place in the North-western Mediterranean basin. BASIL also performs profile measurements of particle backscatter at 355, 532, and 1064 nm, particle extinction at 355 and 532 nm, and particle depolarization at 355 and 532 nm (Di Girolamo et al. 2009b, 2012a, 2012b, 2018b).

BASIL makes use of a Nd:YAG laser source capable of emitting pulses at 355, 532 and 1064 nm, with single pulse energies at 355 nm, i.e. the wavelength used to stimulate rotational and roto-vibrational Raman scattering from atmospheric molecules, of 500 mJ (average optical power of 10 W at a laser repetition rate of 20 Hz). The receiver includes a Newtonian telescope (45-cm diameter primary mirror). Data are sampled with a rough vertical and temporal resolution of 7.5 m and 10 sec, respectively, but vertical and temporal smoothing is typically applied when processing water vapour, temperature and particle backscatter measurements for the purpose of estimating the ABLH. In the present study we considered a vertical and temporal resolution of are 30 m and 5 min, respectively. Based on these vertical and temporal resolutions, water vapour and temperature profile measurements from BASIL extend from the proximity of the surface (50-100 m above station level) up to ~4/~10 km (day/night) and ~6/~20 km(day/night), respectively.

### 3.2 UHF wind profiler

Wind profilers are quite effective and very often used in long-term ABLH measurements as a result of their unattended operation over extended observation periods and the availability of networks of operational wind profilers over wide areas of the globe. However, the operational use of wind profilers is limited by their lack of sensitivity within the surface atmospheric layer (up to 500 m), which prevents from an adequate monitoring of the ABLH and structure at night or in the presence of shallow ABLs.

The five-beam wind profiler (WPR) used in this study was also deployed in Candillargues, approximately 100 m away from the Raman lidar. The system is manufactured by Degreane (model PCL 1300) and operates in the UHF band with a primary frequency at 1.274 GHz. A detailed description of the WPR, its specifications and main working parameters, data processing methodologies and delivered geophysical products is given in Saïd et al. (2016). The WPR operated almost continuously throughout the duration of HyMeX-SOP1 (Saïd et al., 2018). For the purpose of the ABLH measurements reported in this paper, the WPR was operated in low mode, with a pulse length of 1μs, which allows sampling the lower troposphere from 0.15 to 5.7 km a.g.l., with a vertical resolution of 150 m. The methodology applied to determine the ABLH relies on the identification of a distinctive strong peak in the WPR time-height reflectivity plot (Gage et al., 1990), which is associated with turbulence-generated radio refractive index fluctuations, associated atmospheric thermodynamic parameters' fluctuations, though the strength of this peak may depend also on other factors. Wind profiling radars are sensitive to scales

of turbulence that equal half the radar wavelength. At 1.274 GHz, this wavelength is ~20 cm. Therefore the wind profiler is

sensitive to turbulent eddies with spatial dimensions of ~10 cm causing fluctuations in the radio refractive index. In the boundary layer, essentially all scales of turbulence exist from 1-cm wavelengths up. Wind velocity measurements rely on the detection of the Doppler shift, assuming that fluctuations in the radio refractive index, dependent on  atmospheric thermodynamic parameters, are carried along the mean wind flow.

**3.3 Radiosoundings**

A radiosonde launching facility was installed and setup in Candillargues in the proximity of the Raman lidar shortly before the start of HyMeX-SOP 1 in early September 2012. Radiosondes, manufactured by Vaisala (model: RS92), were launched without a predefined schedule, primarily during the intensive observation periods, with a launching rate of up to one launch every 1.5 h. The radiosondes were set to provide vertical profiles of atmospheric pressure, temperature, humidity, and wind

direction and speed during both the ascent and descent phases. For the accuracy of pressure, temperature, humidity, and wind sensors on-board Vaisala RS92 radiosondes the reader should refer to the company datasheet.

**3.4 IGRA DataBase**

ABLH estimates were also inferred from the radiosondes launched from nearest Integrated Global Radiosonde archive

(IGRA) radiosounding station. The specific IGRA station considered in our study is Nimes-Courbessac (lat: 43.8569°N, 4.4064°E, 60 m asl, WMO index= 7645), typically carrying out four radiosonde launches per day. The IGRA radiosondes are characterized by a lower resolution than  the Vaisala RS92 radiosondes. Consequently, data from the two radiosondes have been interpolated on a common height array, with the deviations associated with the interpolation procedure having been verified to be negligible. The launched GPS radiosondes are manufactured by Meteomodem (model: M10). IGRA is

the most comprehensive, authoritative collection of historical and near-real-time radiosonde and pilot balloon observations, with global coverage, maintained and distributed by the National Oceanic and Atmospheric Administration's National Centers for Environmental Information (NCEI). Data were extracted from the IGRA database (version V2), which was released in 2016 (Durré et al., 2018) and includes enhanced quality data with respect to the previous database (version V1). The performance of M10 radiosondes has been assessed and verified during the WMO 2010 radiosonde inter-comparison

effort in Yangjang (Nash et al., 2011, Madonna et al. 2021).

**3.5 ECMWF-ERA5**

ECMWF-ERA5 is the latest reanalysis produced by ECMWF, which includes hourly data on regular latitude-longitude grids, with a 0.25° x 0.25° resolution (Hersbach et al., 2020). Atmospheric parameters are provided at 2 m and at 36 additional

pressure    levels.    ERA5    is    publicly    available    through    the    Copernicus    Climate    Data    Store    (CDS, https://cds.climate.copernicus.eu).

In order to properly carry out the comparisons, the nearest ERA5 grid point to the Raman lidar site was considered. We assume that the representativeness uncertainty associated with the use of the nearest grid-point to be comparable with the uncertainty affecting most interpolation approaches (e.g. kriging, bilinear interpolation, etc.). In general, reanalysis reliability can considerably vary depending on the location, time and selected atmospheric variable (Dee et al., 2016).

## 4. Results

### 4.1 Climatological variability throughout October 2012

In this section we illustrate and discuss the results obtained in the comparison of ABLH estimates obtained from different sensors' measurements (Raman lidar BASIL and radiosondes) and model data (ECMWF-ERA5 analysis) through the application of the Richardson number technique. The inter-comparison effort also includes ABLH measurements from the wind profiler, which rely on the turbulence method, as well as measurements obtained from elastic backscatter lidar signals. We first provide a more climatological assessment, focusing on the evolution of the ABLH throughout the month of October 2012. A separate comparison has been carried out for daytime and night-time cases, at 12:00 UTC and 00:00 UTC, respectively (local time is UTC+02:00 hours in this period of the year). The considered sensors and models are averaged over a half hour time interval centered over the comparison time, i.e. over the interval 11:45-12:15 UTC in daytime and 23:45-00:15 UTC at night. Figure 2 illustrates the time evolution of the ABLH as measured/modeled through the above mentioned sensors/models/approaches, with figure $2a_1$ focusing on daytime cases and figure $2a_2$ on night-time cases. The figure includes six distinct ABLH estimates: ABLHs obtained through the application of the Richardson number method to: i) the Raman lidar data, the radiosonde data (considering separately (ii) radiosondes launched on-site and (iii) radiosondes launched from the closest IGRA station) and (iv) the ECMWF-ERA5 analysis data, (v) ABLHs obtained from wind profiler and (vi) ABLHs obtained from elastic backscatter lidar signals. Results reveal a general good agreement between the six different estimates both in daytime and night-time, all of them revealing the major features associated with ABLH monthly variability. However, percentage differences may be large especially during the night, when the ABL becomes shallow. During the month of October 2012, approximately 60 % of the night-time ABLH values are found to not exceed 300 m. Percentage differences in excess of 100 % are found only 2 times in this period and characterize the comparison of BASIL vs. the on-site radiosondes, with the former of these two sensors having a limited sensitivity below 250 m, as a result of the presence of overlap effects and a blind vertical region.

The Richardson number approach applied to the on-site radiosonde profiles is probably the most reliable approach (lowest bias) and, assuming this approach as bias-free, it can be considered as reference. Figures $2b_1$ and $2c_1$ illustrate the daytime deviations, expressed both in meters and in percentage (%), respectively, between the five reminder ABLH estimates and the one obtained through the application of the Richardson number approach to the on-site radiosonde profiles, while figures $2b_2$ and $2c_2$ illustrate the night-time deviations. These values are also reported in table 2. Most deviation values are within ± 200 m and ± 20 %. Again, larger deviation values are found to characterize those comparisons considering lidar-based estimates of the ABLH, as a result of the above mentioned overlap effects and the presence of a blind vertical region.

The mean bias throughout the month of October 2012 of each ABLH estimate with respect to the reference value has also been computed. The smallest deviations are observed for nigh-time comparisons, with absolute mean biases in the range 18.2-61.6 m and relative mean biases in the range 3.0-26.4 %. More specifically, very smallest biases are found to characterize the ABLH estimates obtained through the application of the Richardson number method (18.2 m /7.4 % for the IGRA radiosondes, 20.5 m/8.15 % for the Raman lidar BASIL, and -28 m/-2.97 % for ECMWF-ERA5 analysis), while slightly large bias values are found to characterize ABLH estimates from the wind profiler (47 m/21.7 %) and ABLH estimates from elastic backscatter lidar signals (61.6 m/26.4 %). The slightly smaller ABLH values characterizing ECMWF-ERA5 analyses are probably to be attributed to the systematically smaller values of the water vapour mixing ratio and the wind V component from ECMWF-ERA5 with respect to those from other sensors. Another possible motivation for the negative systematic bias affecting ABLH estimates from ECMWF-ERA5 is represented by the missed assimilation of the on-site radiosondes in ECMWF-ERA5.

Slightly larger absolute deviations are observed for daytime comparisons, with absolute biases in the range 46-151 m and percentage biases in the range 6.8-17.5 %. Again, smaller biases are found to characterize night-time ABLH estimates obtained through the application of the Richardson number method (46 m/6.76 % for the Raman lidar BASIL, 68 m /9.74 % for the IGRA radiosondes, and 105 m/7.6 % for ECMWF-ERA5 analysis), while larger values are found to characterize ABLH estimates from the wind profiler (151 m/17.5 %) and ABLH estimates from elastic backscatter lidar signals (109 m/14.4 %). In general, there is a negative bias of ERA5, this is because in general we found for parameters such as q and v necessary for the computation of Rib related to comparison of radiosonde on site and the Era5 a max error that is around 50% under extimation for mx and about 20% for wind speed, while temperature are quite irrelevant.

Figure 3 compares the different ABLH estimates in terms of scatter plots. Each scatter plot includes 31 data points, one per day throughout the month of October 2012. Again, the different ABLH estimates are compared both in daytime (panels $a_1$-$e_1$ in figure 3) and at night (panels $a_2$-$e_2$). A linear fit is applied to the data points, using a linear regression function passing through zero, with the form $Y = A \times X$. X are the ABLH reference values and Y are the values from the five reminder ABLH estimates. The term A represents the slope of the fitting line and provides an alternative estimate of the bias of each of the five ABLH estimates with respect to the reference, while the correlation coefficient $R^2$ of the linear fit quantifies the degree of agreement between the compared ABLH estimates. Values of slope and $R^2$ for each ABLH estimate are reported in the table 3a for daytime comparisons and in table 3 b for night time comparisons. All values of $R^2$ are in the range 0.94-0.98, which testifies the high level of agreement between the different ABLH estimates both in daytime and at night, while all values of slope are in the range 0.91-1.08 for daytime comparisons and in the range 0.95-1.03, which testifies the very small bias affecting all five ABLH estimates with respect to the reference estimate. Again, slightly larger biases are observed for daytime comparisons with respect to night-time, this confirming the results already illustrated in the preceding part of the paper.

More specifically, figure $3a_1$ compares with the reference values daytime ABLH estimates obtained through the Richardson number method (RNM) applied to the Raman lidar data, with $R^2$ being equal to 0.98 and slope being equal to 1.02. This

result confirms the small bias (2 %) affecting ABLH estimates from the Raman lidar data when compared with the reference estimates. Identical values of $R^2$ and slope are found in the night-time comparison (figure $3a_2$). Figure $3b_1$ compares with the reference values daytime ABLH estimates obtained through the RNM applied to the IGRA radiosonde data, with $R^2$ being equal to 0.94 and slope being equal to 1.01. This result confirms a very small bias (1 %) affecting ABLH estimates from the IGRA radiosonde data when compared with the reference estimates. Very similar values of $R^2$ and slope, 0.95 and 0.98,

respectively, are found in the night-time comparison (figure $3b_2$). Figure $3c_1$ compares with the reference values daytime ABLH estimates obtained through the RNM applied to the ECMWF-ERA5 analysis data, with $R^2$ being equal to 0.98 and slope being equal to 0.91. This result confirms a small bias (9 %) affecting the ABLH estimate from the ECMWF-ERA5 analysis data when compared with the reference estimate. Values of $R^2$ and slope for the night-time comparison are 0.97 and 0.95, respectively (figure $3c_2$), which confirm the presence of a slightly smaller bias at night with respect to daytime. Figure

$3d_1$ compares daytime ABLH estimates from the wind profiler with the reference ABLH values, with $R^2$ being equal to 0.95 and slope being equal to 1.08. This result confirms a small bias (8 %) affecting the ABLH estimate from the wind profiler when compared with the reference estimate. Values of $R^2$ and slope for the night-time comparison are 0.95 and 1.03, respectively (figure $3d_2$), which confirm the presence of a slightly smaller bias at night with respect to daytime. Finally, figure $3e_1$ compares daytime ABLH estimates obtained from range-corrected elastic backscatter lidar signals with the

reference ABLH values, with $R^2$ being equal to 0.96 and slope being equal to 1.05. This result confirms a small bias (5 %) affecting the ABLH estimate obtained from the range-corrected elastic backscatter lidar signals when compared with the reference estimate. Values of $R^2$ and slope for the night-time comparison are 0.96 and 1.02, respectively (figure $3e_2$), which confirm the presence of a slightly smaller bias at night with respect to daytime.

Results clearly reveal that absolute biases are smaller during the night than during the day. This is most probably due to the

fact that night-time portions of the measurement records are typically characterized by higher stability conditions. However, percentage biases are typically larger during the night, when the ABL may become very shallow. In fact, in the presence of shallow ABLs, ABLH values become comparable with the measurement uncertainty and this makes percentage biases intrinsically high. In our specific case, ABLH values are found to not exceed 300 m throughout most part of the month of October 2012. With a specified vertical resolution of 150 m, the uncertainty affecting the estimate of the ABLH is 50 %

when sounding ABLs with heights not exceeding 300 m. As already anticipated above, the bias affecting the ABLH obtained from the range-corrected elastic backscatter lidar signals is possibly generated by the limited sensitivity of this sensor below 250 m, as a result of the presence of overlap effects and a blind vertical region. Furthermore, the negative biases observed in ERA5 ABLH estimate are most probably associated with the negative bias affecting ERA5 water vapour and wind V component data.

The above results reveal that the different ABLH methods considered in the paper, which refer to different definitions and physical (dynamic and thermodynamic) processes, ultimately lead to ABLH estimates that are in very good agreement among them. This conclusion, which need to be confirmed over longer time series including a complete seasonal cycle, confirms that turbulent air motion and vertical mixing, induced by shear and buoyancy forces (Stull, 1988), cause rapid

fluctuations of several physical quantities, such as flow velocity, temperature, moisture and aerosol concentration, which are

strongly correlated one with the other. This result confirms the correctness of the considered approach to simultaneously apply different ABLH estimation methods referring to the variability of different physical quantities.

Additional parameters from ERA5 reanalysis data have been considered to corroborate the analysis and interpretation of the observed atmospheric features. Figure 4 shows the time evolution of the Convective Available Potential Energy (CAPE, panel a), the friction velocity (panel b) and the relative humidity (panel c) from ERA5 reanalysis over the time period 1-31

October 2012. Friction velocity quantifies the vertical transport of momentum, or turbulence generation. It is calculated as the square root of the surface stress divided by air density. Friction velocity includes both a turbulent and a viscous component. In a turbulent flow, friction velocity is approximately constant within the few lowest meters of the atmosphere. This parameter increases with surface roughness. CAPE quantifies atmospheric instability, instability (i.e. work done by the buoyancy on the air mass), with large positive values (>400) being necessary for the onset of convective activity in a

conditionally unstable tropospheric layer. CAPE is strongly controlled by the ABL properties, with their coupling holding in both convective and non-convective conditions (Donner and Phillips, 2003). Additionally, the response of the boundary layer to relative humidity is investigated although it is known that it involves competing mechanisms (Ek and Mahrt, 1994) and the net effect on relative humidity is difficult to disentangle. Nevertheless, in literature, the effect of relative humidity on the ABLH and its variability is often investigated because, in general, the ABLH is higher for high surface temperature and low

humidity values, which translates into surface sensible heat fluxes dominating over latent heat fluxes and leading to increased buoyancy (Zhang et al., 2013).

Peak values in friction velocity are found in association with the highest ABLH values. During the month of October 2012 this happens around 18-19 October. This result confirms the important role played by the vertical transport of momentum and turbulence in the development of the ABL and in ABLH growth. CAPE values in excess of 100 J kg$^{-1}$ are observed in

the time periods 9-11, 19-21 and 25-27 October 2012, but no evident correlation is found with the corresponding time evolution of the ABLH. Within all other periods CAPE values are in the range 50-100 J kg$^{-1}$, which are indicative of high stability conditions of an atmospheric environment unfavorable to convective activity on these days. Relative humidity values are found to experience a large variability, but no clear correlation appear to be present with ABLH estimates. The correlation between relative humidity and ABLH was verified in order to assess the potential role of water vapor in feeding

convective activity and consequently in determining an increase of the ABLH when triggering mechanisms are present.

The above results reveal the role of atmospheric stability in limiting the variability of the ABLH. The high stability during most of the month of October is probably one of the main drivers of the very good correlation found between the different sensors/models/methods. Figure 5 illustrates the percentage bias of the different ABLH estimates with respect to the one obtained through the application of the Richardson number approach to the on-site radiosonde profiles, expressed as a

function of the atmospheric stability conditions. The presence of stable or unstable conditions was verified based on the use of the "lifted index" obtained from ERA5 model analysis data, with negative values for this index indicating instability - the more negative, the more unstable the air is - and positive values indicating stability. More than 50 % of the cases can be

classified as stable condition, while the reminder cases can be classified as unstable. For the unstable conditions mutual deviations between the different/sensors methods are approximately 30% larger than in the case of stable conditions. A lower dispersion of the bias values implies a higher correlation among the different ABLH estimates in case of stable weather conditions.

We also tried to quantitative correlate ABLH estimates with the variability of CAPE, friction velocity and relative humidity. A linear fit was applied to the time series of CAPE day-by-day gradient, the friction velocity and the relative humidity values in the  time period 1-31 October 2012 vs. the corresponding ABLH estimates (plots not shown here). The correlation between ABLH estimates and the corresponding CAPE day-by-day gradient values is found to be 0.51 in day-time and 0.46 at night; the correlation between ABLH estimates and the corresponding friction velocity values is found to be 0.75 in day-time and 0.91 at night; finally, the correlation between ABLH estimates and the corresponding relative humidity values is found to be 0.41 in day-time and 0.45 at night. These results reveal a reasonably good correlation between ABLH estimates and corresponding friction velocity values and a very mild correlation between ABLH estimates and corresponding CAPE day-by-day gradient and relative humidity values. In order to further underline the correlation between ABLH estimates and corresponding friction velocity values, a scatter plot of ABLH vs. friction velocity-ABLH) for the month of October 2012 is illustrated in figure 6. In order to reveal the higher correlation during night-time, data points are plotted separately for both daytime (red dots, 12:00 UTC) and night-time (black dots, 00:00 UTC).

**4.2 Short-term variability over the two-day period18 and 19 October 2012**

For the purpose of assessing the performance in monitoring the short-term variability of the ABLH, our analysis was also focused on one specific extended case study, covering the two complete daily cycles on 18 and 19 October 2012 and the night in between. Figure 7a illustrates the time-height cross section of the range corrected signals (RCS) at 355 nm as measured by BASIL over the time interval from 09:00 UTC on 18 October 2012 to 19:00 on 19 October 2012. The figure includes approximately 400 consecutive RCS  profiles, each one integrated over a time interval of 5 min and with a vertical resolution of 150 m. The six ABLH estimates are also included in the figure, with different symbols being used to identify the different ABLH sensors/models/approaches.

This measurement period is characterized by variable weather conditions, which translate into variable aerosol and water vapour concentrations at different altitudes, with the presence of both surface and elevated aerosol and humidity layers. Such conditions are ideal to assess the performance of the different sensors/models/approaches and their capability to resolve the short-term variability of the ABLH. Specifically, on 18 October 2012 an elevated Mesoscale Convective System (MCS) is found to transit over the lidar station, with the cloud system base ranging between 2.5 and 4.5 km. Light precipitations are observed around 11:00 and 12 UTC, while virga events, with precipitating particles sublimating before reaching the ground, are observed later in the afternoon in the time interval 17:00-19:00 UTC. On 18 October 2012 the ABLH is found to descend from an initial value of 0.8-1.0 km around 09:30 UTC down to a minimum of 0.5-0.7 km around 18:00 UTC and then keep almost constant until 23:30 UTC. The limited growth of the ABLH on 18 October is probably associated with the shading

effect of clouds, which prevent from an effective onset of convective activity. Stratiform clouds are found to transit over the lidar site in the time interval 02:00-05:00 UTC on 19 October, with strong evidence of them in the RCSs (cloud base is at 1-0-1.2 km and vertical extent is ~0.2 km). ). Starting around 07:00 UTC on 19 October stratiform clouds are continuously present throughout the day, with a cloud base progressively descending from ~1.4 km to ~0.8 km. Evidence of shallow orographic stratiform precipitation is also observed throughout the day. All ABLH estimates indicate a marked increase around 00:00 on 19 October, with values rising from 0.5-0.7 km up to 1.1-1.3 km. Such abrupt increase in ABLH is probably associated with the observed changes in the wind field flow, with the wind direction turning from East to North and the wind speed decreasing from ~20 down 3-5 knots, and the consequent change of sounded air masses. This change in the ABL structure is caused by advection of air masses with different thermodynamic and compositional properties associated with the wind direction turning, ultimately altering the stability and turbulent state of the sounded air masses. A similar abrupt ABLH variability had been reported by Pal and Lee (2019), who revealed the important role played in coastal areas by advection via onshore and offshore flows. All six ABLH estimates are coherent in revealing this abrupt ABLH increase.

Coming to the specific details of the comparison between the different ABLH estimates, throughout the day on 18 October the six estimates are all in very good agreement, with values always within 200 m one from the other, with the only exception of a few data points. Throughout the day on 19 October, despite the potential issues/problems associated the with the presence of the thick stratiform clouds and light precipitation, all six ABLH estimates are in reasonable good agreement, with all values always within 200-300 m. On 19 October ABLH estimates from the wind profiler are systematically found to be slightly smaller than all other estimates. The slightly larger bias of the wind profiler with respect to the other ABLH estimates was already underlined above when focusing on the climatological analysis throughout the month of October 2012. None of the six ABLH estimates appears to be affected by the presence of the thick stratiform clouds, not even the estimates based on the use of the elastic backscatter signals.

Figure 7b illustrates the time-height cross-section of the water vapour mixing ratio measurements carried out by BASIL on the same day, which are displayed over the same time intervals considered in figure 7a. A dry layer appears at the ABL top throughout most part of the day on 18 October, probably resulting from sub-cloud low-level rain evaporation. This dry layer may ultimately have contributed to convection regeneration events observed throughout the passage of the MCS (Li et al. 2009; Morrison et al. 2009). It is to be specified that rain evaporation significantly contributes to the heat and moisture budgets of clouds (Emanuel et al., 1994), but few observations of these processes are available (Gamache et al., 1993). Dry layers are frequently observed in mid-latitude convective environments as a result of air being advected from different source regions under directionally sheared vertical wind profiles (Carlson and Ludlam 1968). Deep convective precipitation events are influenced, and frequently favored, by the presence of aerosols and midlevel dry layers, and this circumstance may have played a significant role in the formation and development of the observed MCS on this day.

## 5. Conclusions

In the present paper we illustrate and discuss the results from an inter-comparison effort considering ABL height estimates obtained from different sensors, models and techniques. The effort was carried out in the framework of HyMeX-SOP1. ABLH estimates were obtained based on the application of the Richardson number technique to Raman lidar and radiosonde measurements and to ECMWF-ERA5 reanalysis data. The inter-comparison also includes ABLH measurements from the wind profiler, which rely on the turbulence method, as well as measurements obtained from elastic backscatter lidar signals.

A climatological assessment focusing on the evolution of the ABL height throughout the duration of the month of October 2012 was provided, with the inter-comparison being extended to both daytime and night-time data.

In the inter-comparison effort the Richardson number approach applied to the on-site radiosonde data is taken as reference. Results reveal a good agreement between the different sensors/models/approaches, all of them being able to capture the major features ABLH time evolutions. Values of the correlation coefficient $R^2$ in the range 0.94 to 0.98. Biases of the single ABLH estimates with respect to the reference ABLH values were also determined through both a simple statistical analysis (mean deviation between the single sensors/model values and the reference values) and a regression analysis (slope of the linear fit regression line correlating the single sensor/model values and the reference values), with all bias values being smaller that 9 % in daytime and smaller than 5 % at night. The analysis was integrated with the comparison with a variety of atmospheric dynamic and thermodynamic variables from ERA 5, namely Convective Available Potential Energy, friction velocity and relative humidity, with friction velocity being found to be one of the main drivers of the ABLH variability.

The analysis was also focused on one specific case study, covering an extended time interval from 09:00 on 18 October 2012 to 19:00 UTC on 19 October 2012, including two daytime portions, the first one characterized by the presence of high scattered clouds between 3 and 4 km and the second one characterized by the present of low stratiform clouds between 1 and 2 km, which allowed to assess the performance in the characterization of the short-term variability of the ABLH in variable weather conditions. This final analysis, while providing preliminary encouraging results, has in no extend to be considered as a thorough demonstration of the applicability of the considered approaches to variable complex weather conditions as in fact a more comprehensive and extensive study has to be carried out in the future in this direction.

## 5. Acknowledgments

Wind Profiler dataset were obtained from the HyMeX program, sponsored by Grants MISTRALS/HyMeX and ANR-11-BS56-0005 IODA-MED project (contact person: Said Frèdèrique Laboratoire d'Aérologie, Université de Toulouse, UMR CNRS 5560). This work was possible based on the support from the Italian Ministry for Education, University and Research under the Grants OT4CLIMA and FISR2019-CONCERNING, and the support of the Italian Space Agency, under the Grant As-ATLAS and CALIGOLA.

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

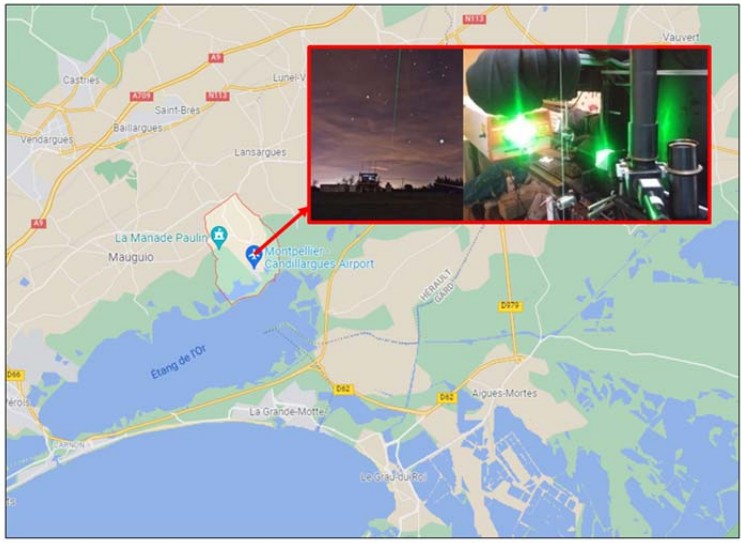

**Figure 1**: Location and two images of the Raman lidar system BASIL operated during HyMex-SOP1 (© Google Maps 2021).

| Instruments Analysis used | PROs | CONs |
|---|---|---|
| Radiosondes | In situ measurements of all variables typically used for the ABLH retrieval; high vertical resolution. | collocation mismatch; sensitivity to applied thresholds in altitude and turbulence; with rain, effect of drops or evaporation cooling on the sensor; radiosonde pendulum motion in the first km affecting wind measurements mainly. |
| Raman lidar | high temporal sampling; high vertical resolution | lack of sensitivity below 250-400 m agl; gradient attribution due to multiple gradients; in some case low SNR may require smoothing; sensitivity to applied thresholds in altitude to limit other gradients; no measurements with rain. (max vertical resolution: 7.5m; max time resolution: 10 sec) |
| UHF wind profiler | High temporal sampling; all weather measurements. | medium-low resolution; lack of sensitivity below 500 m agl; sensitivity to birds and clutter, reducing the detectability of the atmospheric signal on one or more of the off-vertical beams; multiple peaks in signal-to-noise-ratio (SNR) with a consequent attribution problem; precipitation can influences the accuracy of wind measurements depending on intensity and duration of precipitation. |
| ERA5 -Reanalysis | Full temporal coverage; no missing data for the study of diurnal cycle; smoother profile. | gridded data with a consequent representativeness uncertainty; medium-low vertical resolution in the ABL; coarse temporal resolution (1h). |

**Table 1:** PROs and CONs of the considered instruments/reanalysis and their potential sources of bias.

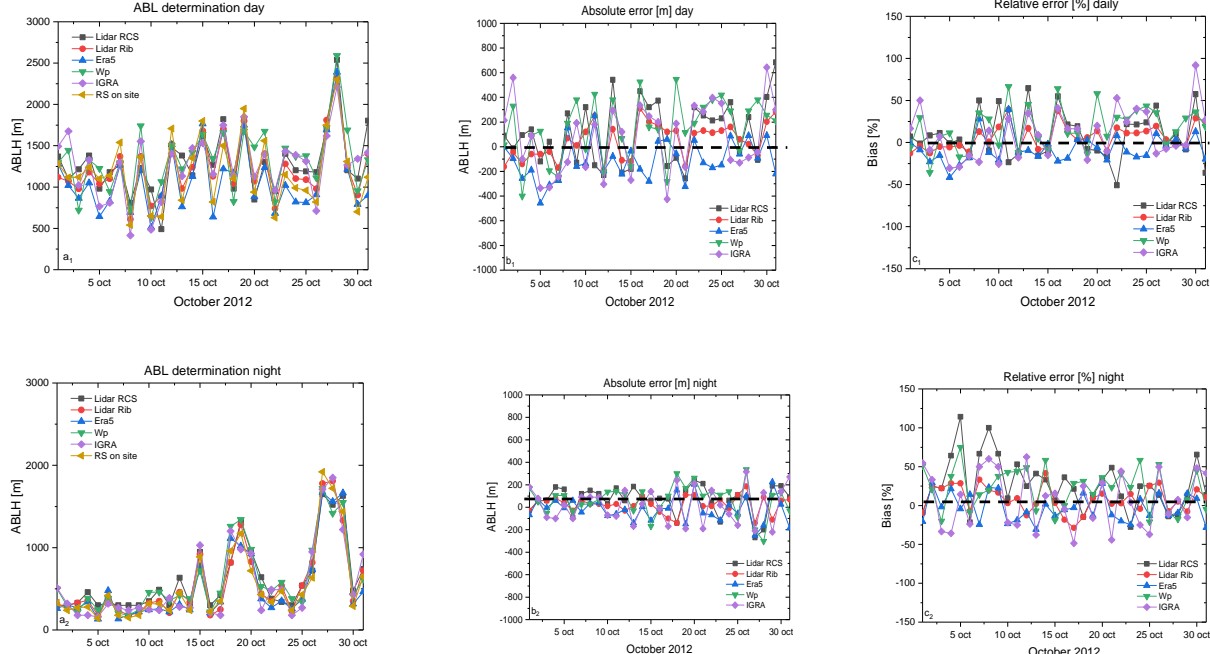

**Figure 2**: Time evolution over the time period 1-31 October 2012 of the six distinct ABLH estimates illustrated in the paper. Panel $a_1$ illustrates the daytime comparison, while panel $a_2$ illustrates the night-time comparison. "Lidar Rib" stands for ABLH estimates obtained through the Richardson number method (RNM) applied to the Raman lidar data, "RS on-site" stands for ABLH estimates obtained through the RNM applied to the on-site radiosonde data, "IGRA" stands for ABLH estimates obtained through the RNM applied to the IGRA
radiosonde data, "ERA5" stands for ABLH estimates obtained through the RNM applied to the ECMWF-ERA5 analysis data, "WP" stands for ABLH estimates obtained from wind profiler, and "Lidar RCS" stands for ABLH estimates obtained from range-corrected elastic backscatter lidar signals. Panels $b_1$-$b_2$ and $c_1$-$c_2$ represent the daytime deviations, expressed both in meters and in percentage (%), respectively, between the ABLH estimate obtained through the application of the Richardson number approach to the on-site radiosonde profiles and the five reminder ABLH estimates. Panels $b_1$ and $c_1$ illustrate the daytime comparison, while panels $b_2$ and $c_2$ illustrate the
night-time comparison.

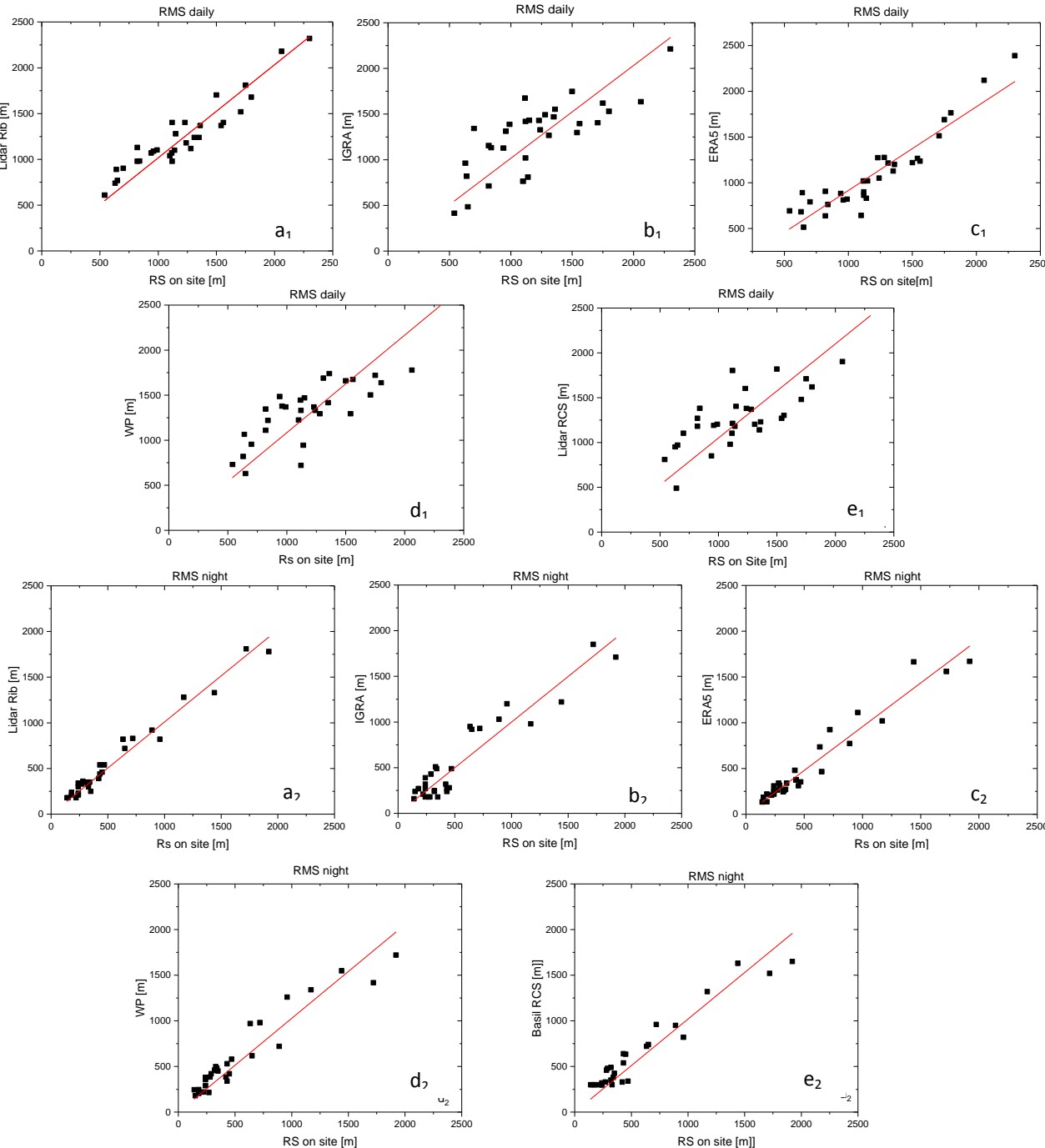

**Figure 3:** Comparison of the different ABLH estimates expressed in terms of scatter plots. On the x-axes of each plot are the ABLH reference values obtained through the application of the Richardson number approach to the on-site radiosonde profiles, while on the y-axes are the values from the five reminder ABLH estimates. Panel a-e represent are for daytime comparisons, panels f-j for night-time. Panel $a_1/a_2$: "Lidar Rib" vs. "RS on-site", panel $b_1/b_2$: "IGRA" vs. "RS on-site", panel $c_1/c_2$: "ERA5" vs. "RS on-site", panel $d_1/d_2$ "WP"
vs. "RS on-site", panel $e_1/e_2$: "Lidar RCS" vs. "RS on-site". The red lines represent the linear fit applied to the data points, using a linear regression function with the form $Y = A \times X$.

| Date | BASIL (RCS) vs RS [12:00] | | BASIL (RCS) vs RS [00:00] | | BASIL Rib vs RS [12:00] | | BASIL Rib vs RS [00:00] | | ERA5 vs RS [12:00] | | ERA5 vs RS [00:00] | | WP vs RS [12:00] | | WP vs RS [00:00] | | IGRA vs RS [12:00] | | IGRA vs RS [00:00] | | St. Dev 12:00 | St. Dev 00:00 |
|---|---|---|---|---|---|---|---|---|---|---|---|---|---|---|---|---|---|---|---|---|---|---|
| | [%] | [m] | [%] | [m] | [%] | [m] | [%] | [m] | [%] | [m] | [%] | [m] | [%] | [m] | [%] | [m] | [%] | [m] | [%] | [m] | [%] | [%] |
| 011012 | 7.03 | 90 | -9.09 | -30 | -12.6 | -162 | -9.09 | -30 | -0.2 | -3 | -20.6 | -68 | -6.22 | -73 | 50.9 | 168 | -16.7 | -213 | -54.5 | -180 | 10.73 | 36.3 |
| 021012 | -1.1 | -13 | 25. | 60 | -4.12 | -46 | 25.0 | 60 | -8.7 | -97 | 29.2 | 70 | 29.4 | 329 | 20.8 | 50 | 50.1 | 559 | -33.3 | -80 | 25.55 | 4.8 |
| 031012 | 8.3 | 94 | 22.3 | 60 | -12.5 | -140 | 22.2 | 60 | -22. | -257 | -1.9 | -5 | -35.7 | -400 | -20.0 | -54 | -8.93 | -100 | 33.3 | 90 | 16.43 | 24.9 |
| 041012 | 11.2 | 140 | 64.3 | 180 | -4.84 | -60 | 28.6 | 80 | -15.1 | -189 | 21.4 | 60 | 7.26 | 90 | 37.5 | 105 | 7.06 | 88 | -35.7 | -100 | 10.95 | 36.7 |
| 051012 | -0.9 | -120 | 114 | 160 | -5.45 | -60 | -28.6 | -40 | -41.5 | -457 | -4.3 | -6 | 11.3 | 125 | -45.0 | -65 | -30.4 | -335 | 14.3 | 20 | 20.89 | 48.3 |
| 061012 | 3.51 | 40 | -21.4 | -90 | -3.51 | -40 | -7.14 | -30 | -27.2 | -310 | 14.3 | 60 | -17.1 | -195 | -8.3 | -35 | -28.9 | -330 | -23.8 | -100 | 14.33 | 15.2 |
| 071012 | -17.5 | -270 | 66.6 | 120 | -11.0 | -170 | -33.3 | -60 | -17.8 | -274 | -24.4 | -44 | -15.9 | -246 | 13.9 | 25 | -15.7 | -243 | 50.0 | 90 | 2.72 | 35.2 |
| 081012 | -25.00 | 270 | 100 | 150 | 12.9 | 70 | 20.1 | 30 | 28.1 | 152 | 23.3 | 35 | 35.2 | 190 | 20.0 | 30 | -23.2 | -125 | 60.0 | 90 | 27.87 | 35.2 |
| 091012 | -9.56 | -130 | 66.5 | 120 | 0.74 | 10 | 16.7 | 30 | -11.7 | -160 | 22.2 | 40 | 27.9 | 380 | -37.8 | -68 | 14.2 | 193 | 50.0 | 90 | 16.72 | 20.4 |
| 101012 | 49.23 | 320 | 9.38 | 30 | 18.4 | 120 | 3.13 | 10 | -20.9 | -136 | -23.4 | -75 | -3.08 | -20 | 42.8 | 137 | -25.4 | -165 | -21.9 | -70 | 30.78 | 27.1 |
| 111012 | -23.4 | -150 | 53.1 | 170 | 39.0 | 250 | 9.38 | 30 | 39.3 | 252 | -18.8 | -60 | 66.4 | 425 | 44.1 | 141 | 28.1 | 180 | -25.0 | -80 | 32.99 | 35.5 |
| 121012 | -13.4 | -229 | 25.1 | 60 | -11.1 | -189 | -12.5 | -30 | -11.4 | -196 | -8.3 | -20 | -12.0 | -207 | 48.8 | 117 | -17.8 | -304 | 32.5 | 88 | 2.73 | 33.4 |
| 131012 | 64.52 | 542 | 41.1 | 185 | 16.9 | 142 | 2.22 | 10 | -9.29 | -78 | -30.9 | -139 | 45.2 | 380 | -6.7 | -30 | 34.8 | 293 | -37.8 | -170 | 28.11 | 31.3 |
| 141012 | -15.6 | -210 | 33.3 | 80 | -8.15 | -110 | 41.7 | 100 | -16.4 | -222 | 1.7 | 4 | 5.00 | 68 | 58.3 | 140 | 8.89 | 120 | 12.5 | 30 | 11.67 | 22.7 |
| 151012 | -10.1 | -180 | 6.74 | 60 | -6.67 | -120 | 3.37 | 30 | -1.94 | -35 | -13.1 | -117 | -9.00 | -162 | -19.1 | -170 | -15.0 | -270 | 15.7 | 140 | 4.77 | 14.4 |
| 161012 | 54.88 | 450 | 36.3 | 80 | 37.8 | 310 | -18.2 | -40 | -22.2 | -182 | -5.5 | -12 | 34.0 | 325 | -8.5 | 143 | 40.9 | 335 | -4.5 | -10 | 33.73 | 20.6 |
| 171012 | 21.33 | 320 | 21.4 | 75 | 13.5 | 203 | -28.6 | -100 | -18.6 | -280 | -2.9 | -10 | 10.6 | 160 | 28.6 | 100 | 16.5 | 248 | -48.6 | -170 | 15.78 | 32.7 |
| 181012 | 19.33 | 170 | -14.5 | -140 | 3,1 | 40 | -14.6 | -140 | -5.7 | -64 | 15.8 | 152 | 12.3 | 120 | -31.3 | -300 | 9.2 | 110 | -14.0 | -120 | 9.84 | 21.8 |
| 191012 | -7.62 | -157 | -2.24 | -40 | 5.83 | 120 | 0.51 | 10 | 2.91 | 60 | -11.8 | -130 | -13.7 | -283 | -31.3 | 630 | -20.6 | -425 | -7.62 | -150 | 11.11 | 13.8 |
| 201012 | -9.57 | -90 | 33.3 | 240 | 13.8 | 130 | 15.3 | 110 | -6.06 | -57 | 28.3 | 204 | 57.9 | 545 | 36.1 | 260 | 19.9 | 188 | 29.2 | 210 | 27.02 | 8.0 |
| 211012 | -16.5 | -257 | 48.8 | 210 | -10.0 | -157 | 2.33 | 10 | -20.7 | -323 | -12.1 | -52 | 7.37 | 115 | 23.3 | 100 | -10.5 | -165 | -44.2 | -190 | 10.71 | 35.2 |
| 221012 | -50.79 | -320 | 11.7 | 40 | 17.4 | 110 | 2.94 | 10 | 8.25 | 52 | -20.0 | -68 | 30.1 | 190 | 41.2 | 140 | 52.7 | 333 | 44.1 | 150 | 19.77 | 27.0 |
| 231012 | 22.00 | 253 | -27.6 | -130 | 11.3 | 130 | 14.9 | 70 | -11.3 | -130 | -24.7 | -116 | 27.8 | 320 | 23.4 | 110 | 24.5 | 283 | 4.3 | 20 | 15.89 | 23.1 |
| 241012 | 21.52 | 213 | 25.3 | 60 | 11.4 | 113 | -4.17 | -10 | -17.1 | -170 | 8.3 | 20 | 38.4 | 380 | 58.3 | 140 | 40.1 | 398 | -25.0 | -60 | 23.43 | 31.5 |
| 251012 | 23.96 | 230 | 25.5 | 110 | 13.5 | 130 | 25.5 | 110 | -15.4 | -148 | -13.0 | -56 | 43.5 | 418 | -20.9 | -90 | 36.7 | 353 | -37.2 | -160 | 23.14 | 28.4 |
| 261012 | 43.90 | 360 | 13.5 | 86 | 19.5 | 160 | 29.3 | 150 | 10.6 | 87 | 15.9 | 101 | 35.3 | 290 | 53.0 | 336 | -13.1 | -108 | 49.8 | 316 | 22.30 | 18.5 |
| 271012 | -2.29 | -40 | -14.1 | -270 | 3.43 | 60 | -7.29 | -140 | -3.43 | -60 | -13.0 | -250 | -1.71 | -30 | -10.4 | -200 | -7.43 | -130 | -10.9 | -210 | 3.90 | 2.6 |
| 281012 | 10.43 | 240 | -11.6 | -200 | 0.87 | 20 | 5.23 | 90 | 3.91 | 90 | -9.3 | -160 | 12.6 | 290 | -17.6 | -303 | -3.80 | -88 | 7.6 | 130 | 6.76 | 11.0 |
| 291012 | -8.17 | -107 | 13.1 | 190 | -5.34 | -70 | -7.64 | -110 | -7.18 | -94 | 15.6 | 225 | 29.0 | 380 | 7.5 | 108 | -3.24 | -43 | -15.3 | -220 | 15.76 | 13.5 |
| 301012 | -17.5 | -273 | 65.5 | 190 | 29.0 | 203 | 20.7 | 60 | 13. | 91 | 9.0 | 26 | 36.4 | 255 | 44.8 | 130 | -91.8 | -643 | 48.3 | 140 | 30.4 | 22.6 |
| 311012 | 35.9 | 342 | -23,2 | -72 | 25.3 | 283 | 10.7 | 70 | -19.7 | -220 | 8.3 | -24.5 | 18.7 | 210 | -4.9 | -32 | 26.8 | 300 | -41.5 | -270 | 28.71 | 25.8 |
| **Mean** | **14.4** | **109** | **26.4** | **61.6** | **6.76** | **46** | **8.15** | **20.5** | **-7.6** | **-105** | **-2.97** | **-28** | **17.5** | **151** | **21.7** | **47** | **9.74** | **68** | **7.4** | **18.2** | **17.92** | **24.5** |

**Table 2:** Biases, expressed both in meters and in percentage (%), respectively, of the different ABLH estimates with respect to the one obtained through the application of the Richardson number approach to the on-site radiosonde profiles. The data cover the month of October 2012 and are separately listed for daytime (12:00 UTC) and night-time (00:00 UTC) comparisons.

| a) [12:00 UTC] Summary-FIT Table [Y = A · X] | | | | b) [00:00 UTC] Summary-FIT Table [Y = A · X] | | | |
|---|---|---|---|---|---|---|---|
| omparison Methods | Slope | | Correlation coefficient | Comparison Methods | Slope | | Correlation coefficient |
| RS on site | A | σ(A) | $R^2$ | RS on site | A | σ(A) | $R^2$ |
| Basil (RCS) | 1.05 | 0.038 | 0.96 | Basil (RCS) | 1.02 | 0.035 | 0.96 |
| Basil (Rib) | 1.02 | 0.024 | 0.98 | Basil (Rib) | 1.02 | 0.022 | 0.98 |
| ERA5 | 0.91 | 0.022 | 0.98 | ERA5 | 0.95 | 0.026 | 0.97 |
| Wind-Profiler | 1.08 | 0.038 | 0.95 | Wind-Profiler | 1.03 | 0.038 | 0.95 |
| IGRA-DB | 1.01 | 0.04 | 0.94 | IGRA-DB | 0.98 | 0.035 | 0.95 |

**Table 3:** Results from the regression analysis, with values of the correlation coefficient $R^2$ and the regression line slope A, with its standard deviation σ(A). Results are reported for both daytime (panel a, 12:00 UTC) and night-time (panel b, 00:00 UTC) comparisons.

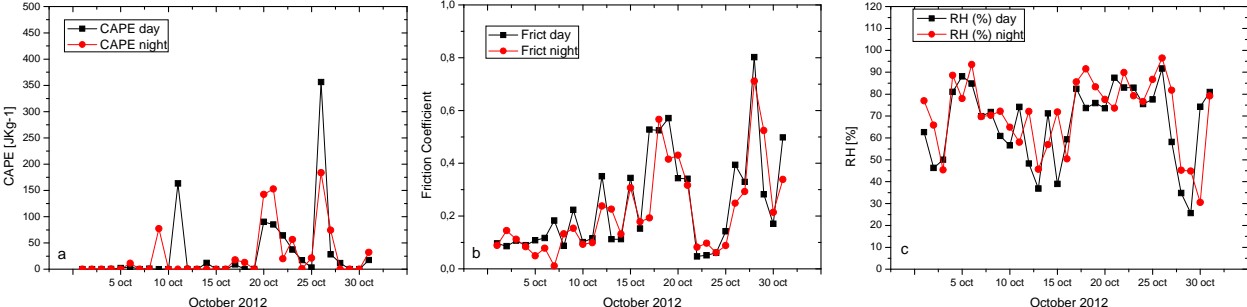

**Figure 4:** ERA5 atmospheric reanalysis for CAPE (panel a). friction velocity (panel b) and relative humidity (panel c) for the month of October 20212. Results are reported for both daytime (black lines, 12:00 UTC) and night-time (red lines, 00:00 UTC) comparisons.

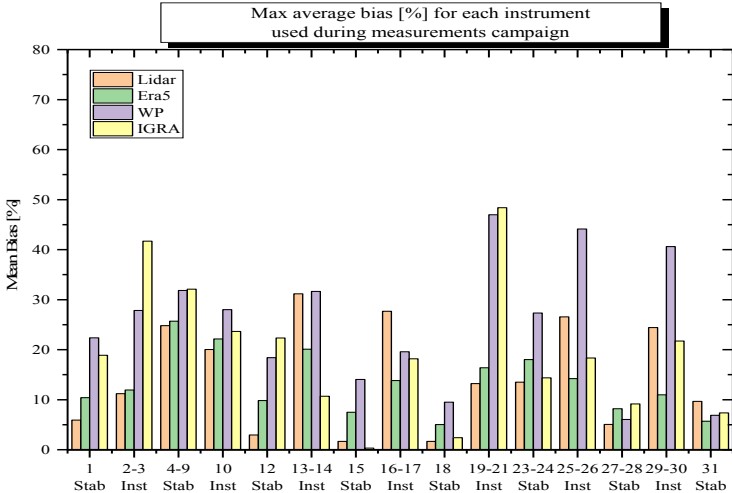

**Figure 5**: Percentage bias of the different ABLH estimates with respect to the one obtained through the application of the Richardson number approach to the on-site radiosonde profiles, expressed as a function of the atmospheric stability conditions.

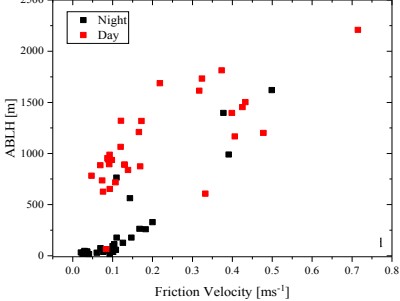

**Figure 6:** Scatter plot of ABLH vs. friction velocity for the month of October 2012. Results are reported for both daytime (red dots, 12:00
UTC) and night-time (black dots, 00:00 UTC).

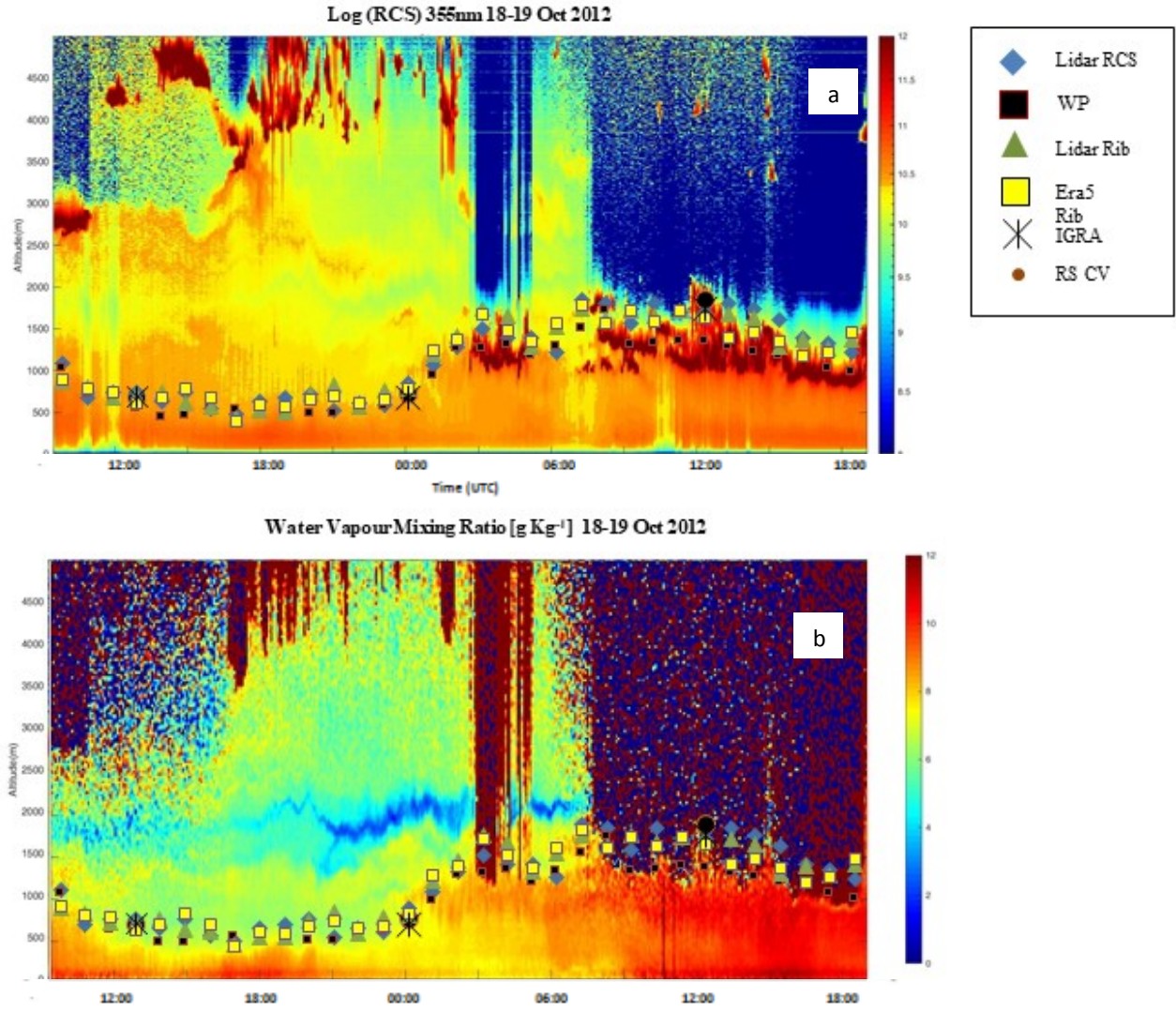

**Figure 7**: (a) Time-height cross section of the Range Corrected signals (RCS) at 355nm (b) and water vapour mixing ratio and as measured by BASIL over the time interval from 23:00 UTC on 18 October 2012 to 19:00 on 19 October 2012. The figure also includes the different ABLS estimates.
