# Peer review of "Inter-comparison of ABL height estimates from different profiling sensors and models in the framework of HyMeX-SOP1"

_Atmospheric Measurement Techniques, 2021_

## Author Comment (AC1)

Dear Editor,

We are very grateful to the two referees for their appropriate and constructive suggestions and for their proposed corrections. We have addressed all issues raised and have modified the paper accordingly. We have also submitted a revised version of the paper where all these changes have been incorporated. We believe that, thanks to precious inputs from the referees, the quality of the manuscript has sensitively improved. Below is a summary of the changes we made and our specific responses to the referees' comments and recommendations.

**Summary of the changes**
**(in black is the original comments of the referee and in red our responses)**

**Referee #1**

The study compare the atmospheric boundary layer height determined by radio-sounding temperature profile, IGRA, wind profiler, ERA5 model, BASIL raman lidar backscattering profile. The description of the ABL structure is lacking detailed information, e.g. the term ABLH is use independently of the instrument and method used, even they are referring to different ABL sublayers.

It is indeed true that the term ABLH had been erroneously used to refer to different heights identified with different instruments and methods. This aspect has been now drastically improved (see more specific comments on this point below).

Strong methodological problems further invalidate the found results.

The strong methodological problems invalidating the results have been either clarified (First methodology problem) or solved (Second methodology problem).

**Main comments:**

- The notion of ABL height is used in the whole introduction and attributed to several "heights" measured by various instruments and methods. The authors should really attribute the right ABL (sub)structure to the right layer height detection. For example, the temperature inversion (usually used for nocturnal boundary layer detection), the MLH detected by the bulk Richardson method and the LLJ cannot be assimilated to the same ABL substructure. A revision of all the concepts introduced in the introduction and of the use of these concepts through all the paper is necessary.

It is indeed true that a more careful effort was needed to properly finalize the inter-comparison effort as in fact comparing different sensors using different methodologies, which refer to different definitions of the ABLH is certainly wrong. A strong revision effort was put together to improve the paper in the direction of making the results of the present paper more scientifically founded. Now, a single approach has been applied to all sensors/models, as long as this has been possible. More specifically, ABLH estimates are obtained based to the application of the Richardson number approach i) to the Raman lidar measurements, to the radiosonde measurements, both ii) those launches on-site and iii) those launched from the nearby IGRA station, iv) to the ECMWF-ERA5 reanalysis data. The inter-comparison effort also includes v) ABLH estimates from the wind profiler, which rely on the turbulence method, as well as vi) ABLH estimates obtained from elastic backscatter lidar signals. This approach was considered following the suggestion from the referee to adopt a common single and well recognized methodology (Richardson number approach). In the present inter-comparison effort the Richardson number approach applied to the on-site radiosonde data is taken as reference, again following the suggestion of the referee in this direction.

It is to be specified that the application of the Richardson number method to the wind profiler data and the elastic backscatter lidar signals was not possible as in fact the estimate of the Richardson number requires information on both wind and thermodynamic profiles, which are not available from the wind profiler and the elastic backscatter lidar signals. Consequently, the Richardson number method is applied only to the on-site radiosonde, to the IGRA radiosonde data and to the ECMWF-ERA5 reanalysis data, and different methodologies were applied to the wind profiler data (turbulence method) and the elastic backscatter lidar signals (particle backscatter gradient approach). Additionally, the application of the Richardson number approach to the Raman lidar refers to thermodynamic profile measurements from this sensor and wind measurements from the simultaneous and co-located radiosondes, as the wind measurements are not available from the Raman lidar (see more details below).

As a results of these methodological changes, large portions of section 1 (Introduction), section 2 (Methods considered for the determination of the ABLH), section 3 (Profiling sensors and model data involved in the inter-comparison effort) and section 4 (Results) have been substantially re-written.

- First methodology problem: the ABL is subjected to a diurnal cycle that is clearly described in case of fair-weather day by Stüll (1989). The authors chose to use the mean ABLH over the entire daytime (from sunrise to sunset) to compare the instruments and methods.

Here there was, and we really have to apologize for that, a miscommunication among the authors. The author who finally reviewed the paper (Paolo Di Girolamo) had not properly interpreted the information coming from the other coauthors taking care of the data analysis. In fact, the ABLH is not estimated as a mean over the entire day, but a much shorter time interval (half hour) is considered. The considered approach was to concentrate on the specific times when the RS data were available, typically at 00:00 and 12:00 UTC. So, all considered sensors and models are typically averaged over a half hour time interval centered on the comparison time (00:00 UTC and 12:00 UTC), i.e. 23.45-00:15 UTC for night-time comparisons and 11:45-12:15 UTC for daytime comparisons.

As a results of these changes, the first paragraph of section 4 (Results) has been completely re-written and the following new text has now been introduced: "In this section we illustrate and discuss the results obtained in the comparison of ABLH estimates obtained from different sensors' measurements (Raman lidar BASIL and radiosondes) and model data (ECMWF-ERA5 analysis) through the application of the Richardson number technique. The inter-comparison effort also includes ABLH measurements from the wind profiler, which rely on the turbulence method, as well as measurements obtained from elastic backscatter lidar signals.

We first provide a more climatological assessment, focusing on the evolution of the ABLH throughout the month of October 2012. A separate comparison has been carried out for daytime and night-time cases, at 12:00 UTC and 00:00 UTC, respectively (local time is UTC+02:00 hours in this period of the year). The considered sensors and models are averaged over a half hour time interval centered over the comparison time, i.e. over the interval 11:45-12:15 UTC in daytime and 23:45-00:15 UTC at night. Figure 2 illustrates the time evolution of the ABLH as measured/modeled through the above mentioned sensors/models/approaches, with figure 2a1 focusing on daytime cases and figure 2a2 on night-time cases. The figure includes six distinct ABLH estimates: ABLHs obtained through the application of the Richardson number method to: i) the Raman lidar data, the radiosonde data (considering separately (ii) radiosondes launched on-site and (iii) radiosondes launched from the closest IGRA station) and (iv) the ECMWF-ERA5 analysis data, (v) ABLHs obtained from wind profiler and (vi) ABLHs obtained from elastic backscatter lidar signals. Results reveal a good agreement between the six different estimates both in daytime and night-time, all of them revealing the major features associated with ABLH monthly variability.

The Richardson number approach applied to the on-site radiosonde profiles is probably the most reliable approach (lowest bias) and, assuming this approach as bias-free, it can be considered as

reference. Figures 2b1 and 2c1 illustrate the daytime deviations, expressed both in meters and in percentage (%), respectively, between the five reminder ABLH estimates and the one obtained through the application of the Richardson number approach to the on-site radiosonde profiles, while figures 2b2 and 2c2 illustrate the daytime deviations. These values are also reported in table 1."

Moreover, this average is done without consideration of cloud cover, precipitation or different state of the atmospheric stability. This impeded the authors to identify potential artifacts of the measurements and of the modeled ABLH, e.g. too high or low ABLH maxima, attribution of ABLH to the cloud base, wrong timing of the ABL maxima.

The reviewer is right in indicating that ABLH estimates in cloudy conditions can potentially be affected by artifacts. A false attribution of the ABLH to a cloud layer is a potential risk, especially when the range corrected signal (RCS) approach based on the detection of particle backscatter gradients is applied. In fact an erroneous identification of the top of the mixed aerosol layer (with aerosol acting as dynamical tracers) can happen when single or multiple cloud layers are present. However it is to be further specified that only the RCS approach can be potentially affected by my cloud contaminations as in fact all other ABLH estimates are not. In this regard it is to be underlined that 4 out of the 6 ABLH estimates illustrated the in the present of paper are the result of the application of the Richardson number method to i) the Raman lidar thermodynamic profiles, ii) to the on-site radiosonde profiles, iii) to the IGRA radiosonde profiles, and and iv) to the ECMWF-ERA 5 reanalysis profiles. The Richardson number method requires information on both wind and thermodynamic profiles, as in fact the estimate of the Richardson number is based on the knowledge of the horizontal wind-speed components' profiles, and $u^2(z)_z$ and $v^2(z)$, and the humidity and temperature profile measurements, needed to determine the virtual potential temperature profile. None of these measurements are affected by the presence of cloud layers. The reminder ABLH estimate considered in the paper comes from the wind profiler data, which are not affected by clouds.
Furthermore, figure 5 has been completely reformulated, with the introduction of the much longer time interval (now from 09:00 UTC on 18 October 2012 to 19:00 UTC on 19 October 2012, while formerly it was from 09:26 to 21:11 UTC on 18 October 2012) and the inclusion of the six selected ABLH estimates. This time interval includes segments with multiple aerosol and cloud layers. Indeed, within this time segment, all six ABLH estimates appear to be in reasonable good agreement, despite the presence of several aerosol and cloud layers. To comment these new results, the following new sentences have been introduced in the text: "Throughout the day on 18 October the six ABLH estimates are all in very good agreement, with values always within 200 m one from the other, with the only exception of a few data points. Throughout the day on 19 October, despite the potential issues/problems associated the with the presence of the thick stratiform clouds and light precipitation, all six ABLH estimates are in reasonable good agreement, with all values always within 200-300 m. On 19 October ABLH estimates from the wind profiler are systematically found to be slightly smaller than all other estimates. None of the six ABLH estimates appears to be affected by the presence of the thick stratiform clouds, not even the estimates based on the use of the elastic backscatter signals."

- Second methodology problem: the authors chose to use the mean of all detection methods (including the measurements and the model) as a reference to estimate the bias of the individual ABLH estimation. This approach does not allow any clear assumptions about the accuracy of the ABLH estimations. Usually the parcel method or the bulk Richardson method applied to the radio-sounding profiles are taken as a reference due to the accuracy of the in-situ measurements.

The approach of estimating the bias with respect to the meaning value of all sensors and approaches is valid only in case a higher reliability cannot be attributed to any of the sensors/approaches. The referee is indeed right in underlying that the Richardson number approach applied to the radiosonde

in situ measurements is probably the most reliable approach, with the lowest bias, and so we can consider this approach as bias-free. All computations and analyses in the paper have now been reformulated considering the deviations and biases of the different sensors/approaches with respect to the ABLH estimates obtained from the application of the Richardson number approach to the on-site radiosonde profiles, which is considered as reference. This change of approach, undertaken following the suggestions from the reviewer, also drastically improved the quality of the comparisons. All linear regression analyses reported in the paper have now improved both in terms of the correlation coefficients $R^2$, with all values now in the range 0.94-0.98, which testifies the high level of agreement between the different ABLH estimates both in daytime and at night, while all values of the slope of the fitting line A are in the range 0.91-1.08 for daytime comparisons and in the range 0.95-1.03, i.e. all values closer to unity, which testifies the very small bias affecting all five ABLH estimates with respect to the reference AHBL estimate.

All the above aspects are now clearly specified in the text, where, among others, the following sentences have been introduced: "The Richardson number approach applied to the on-site radiosonde profiles is probably the most reliable approach (lowest bias) and, assuming this approach as bias-free, it can be considered as reference. Figures 2b1 and 2c1 illustrate the daytime deviations, expressed both in meters and in percentage (%), respectively, between the five reminder ABLH estimates and the one obtained through the application of the Richardson number approach to the on-site radiosonde profiles, while figures 2b2 and 2c2 illustrate the daytime deviations. These values are also reported in table 1." Additionally, the following sentences have been introduced: "Values of A and $R^2$ for each ABLH estimate are reported in the table 2a for daytime comparisons and in table 2 b for night time comparisons. All values of $R^2$ are in the range 0.94-0.98, which testifies the high level of agreement between the different ABLH estimates both in daytime and at night, while all values of A are in the range 0.91-1.08 for daytime comparisons and in the range 0.95-1.03, which testifies the very small bias affecting all five ABLH estimates with respect to the reference AHBL estimate. Again, slightly larger biases are observed for daytime comparisons with respect to night-time, this confirming the results already illustrated in the preceding part of the paper."

- Moreover, the first analysis shows that ERA5 has the worst results, so that it is removed out of the mean of all methods for the second part of the analysis. It is even not clear if EAR5 is removed for the whole month of October or only during the second half of the month when its results strongly differ from the measurements.

In the revised version of the paper we are no longer making use of the ABLH products directly generated by the ECMWF-ERA5 reanalysis, but we are determining the ABLH by applying the Richardson number approach directly to wind and thermodynamic profile analysis data. The present use of the reanalysis data leads to a ABLH estimate which is in much higher agreement with the other five ABLH estimates. As a result of this modification, all ABLH estimates from ECMWF-ERA5 reanalysis data are used and none of them has been removed. Furthermore, the mean ABLH of all methods is no longer used as reference, but the ABLH estimate obtained from the application of the Richardson number approach to the on-site radiosonde profiles, so also this potential source of bias has been removed. The description of the ABLH results from ECMWF-ERA5 reanalysis data has been reformulated in the text and all reported possible motivations for the miss-agreement between ABLH estimated from ECMWF-ERA5 reanalysis data and the other five ABLH estimates have been completely removed from the paper.

All the above aspects are now clearly specified in the text. The following sentences have been introduced in the Abstract: "ABLH estimates were obtained based on the application of the Richardson number technique to Raman lidar and radiosonde measurements and to ECMWF-ERA5 reanalysis data."

The following sentences have been introduced in the Introduction: "In the present research effort we compare ABLH measurements obtained through the application of the Richardson number technique to a variety of sensors and model data, namely the Raman lidar BASIL, radiosondes and ECMWF-

ERA5 analysis data. These results are also compared with ABLH measurements from the wind profiler and from elastic backscatter lidar signals.

Again later in the Introduction it is specified: "In the present paper ABLH estimates obtained through the application of the Richardson number technique to two sensors (Raman lidar and radiosondes) and to the ECMWF-ERA5 model reanalysis data are compared with ABLH measurements from the wind profiler and from elastic backscatter lidar signals."

The following sentences have been introduced in the sub-section 2.1 dedicated to the illustration of the Richardson number method: "Vertical profiles of $T(z)$, $P(z)$ and $\chi_{H2O}(z)$ are available from all sensors/models involved in the inter-comparison effort, namely the Raman lidar, the radiosondes launched on-site and those and from the closest IGRA radiosonde station and the ECMWF-ERA5 analysis data, with the only exception of the wind profiler. The vertical profiles of the horizontal wind-speed components $u(z)$ and $v(z)$, which are needed to quantify the bulk Richardson number are also available from the same sensors/models, with the only exception of the Raman lidar, which only provides the humidity and temperature profile measurements needed to determine the virtual potential temperature profile. In this case, the computation of the Richardson number is completed with the inclusion of wind-speed profile measurements from the simultaneous and co-located radiosondes launched in Candillargues."

Among others, the following sentences have been introduced in section 4 (Results): "In this section we illustrate and discuss the results obtained in the comparison of ABLH estimates obtained from different sensors' measurements (Raman lidar BASIL and radiosondes) and model data (ECMWF-ERA5 analysis) through the application of the Richardson number technique."

- The parcel and bulk Richardson methods could have been applied to the Raman Lidar data allowing a real comparison between the radio-sounding, the model and the lidar ABLH detection. This approach was however not applied by the author.

This has now been done. The bulk Richardson methods has been applied to the Raman Lidar data. As already specified above, in this case the vertical profiles of the horizontal wind-speed components u/(z) and v(z), which are needed to quantify the bulk Richardson number, are not available from the the Raman lidar, which only provides the humidity and temperature profile measurements needed to determine the virtual potential temperature profile. In this case, the computation of the Richardson number is completed with the inclusion of wind-speed profile measurements from the simultaneous and co-located radiosondes launched in Candillargues. This aspect is extensively described in the text. Already in the Introduction, the following sentences have been introduced: "As the estimate of the Richardson number requires information on both wind and thermodynamic profiles, the application of this approach to the Raman lidar relies on thermodynamic profile measurements from this sensor and wind measurements from the simultaneous and co-located radiosondes, as the wind measurements are not available from the Raman lidar." In sub-section 2.1 dedicated to the illustration of the Richardson number method the following sentences have been introduced: "The vertical profiles of the horizontal wind-speed components $u(z)$ and $v(z)$, which are needed to quantify the bulk Richardson number are also available from the same sensors/models, with the only exception of the Raman lidar, which only provides the humidity and temperature profile measurements needed to determine the virtual potential temperature profile. In this case, the computation of the Richardson number is completed with the inclusion of wind-speed profile measurements from the simultaneous and co-located radiosondes launched in Candillargues."

- Figure 5: ABLH corresponding to the maximal gradient of aerosol does not at all corresponds to the red points but is visible e.g. at about 3000 m between 10:30 and 16:00. I then concluded that the used algorithm applied to the raman lidar is not valid.

Figure 5 has been completely reformulated, with the introduction of the much longer time interval (now from 09:00 UTC on 18 October 2012 to 19:00 UTC on 19 October 2012, while formerly it was from 09:26 to 21:11 UTC on 18 October 2012). This time interval includes segments with multiple aerosol and cloud layers. Indeed, within this time segment, all six ABLH estimates appear to be in reasonable good agreement. Results from figure 5 are specifically used to underline the strength and weakness points of the different approaches. The following new sentences have been introduced in the text: "Throughout the day on 18 October the six ABLH estimates are all in very good agreement, with values always within 200 m one from the other, with the only exception of a few data points. Throughout the day on 19 October, despite the potential issues/problems associated the with the presence of the thick stratiform clouds and light precipitation, all six ABLH estimates are in reasonable good agreement, with all values always within 200-300 m. On 19 October ABLH estimates from the wind profiler are found to be systematically slightly smaller than all other estimates. None of the six ABLH estimates appears to be affected by the presence of the thick stratiform clouds, not even the estimates based on the use of the elastic backscatter signals."

- The paper is moreover not well written and structured. A lot of elements are not necessary and some descriptions does not allow the reading to understand the methodology (e.g. if the parcel or the bulk Richardson is used in ERA5).

It is indeed true that the paper was previously not well written and structured. We realized that a lot of elements were not necessary and some descriptions were not effective and did not allow to properly understand the applied methodologies. We went through a complete rewriting and reshuffling of a large portion of the paper. In this direction the paper has also been severely shortened. Almost 4-5 complete pages of old text have been removed from the manuscript.

- lines 44-45: "Specifically, potential temperature tends to keep nearly constant with height within the mixed layer." This is not true since the parcel method used the variation of the potential temperature to determine the convective boundary layer

This sentence has now been removed from the paper, together with the paragraph were it was embedded. In fact the paragraph was dedicated to the description of the ABLH estimate approach based on the identification of local maxima in potential temperature vertical gradient profiles, i.e. the temperature gradient method, which is no longer used in the present research effort.

- lines 45-48: " The level of maximum potential temperature vertical gradient identifies the transition from a convectively unstable region to a stable or more stable region": I've never seen such a definition.

This incorrect sentence has been removed from the paper, together with the paragraph were it was embedded paragraph. In fact the paragraph was dedicated to the description of the ABLH estimate approach based on the identification of local maxima in potential temperature vertical gradient profiles, i.e. the temperature gradient method, which is no longer used in the present research effort.

- Line 54-56: to my knowledge, wind profilers are very often used to detect ABLH and their network is not the denser one.

It is indeed true that wind profilers are very often used to detect ABLH. The corresponding sentence was changed as follows: "Wind profilers are quite effective and very often used in long-term ABLH measurements as a result of their unattended operation over extended observation periods and the availability of networks of operational wind profilers over wide areas of the globe."

- Wind profilers are impacted by birds migration but I never heard about artifacts due to insects' swarms

We learnt about this problem with insects' swarms based on a recent corporation with a research group having a long-term experience in developing and operating wind profilers. We refer to the research group which is leaded by Prof. Frédérique Saïd at Université Toulouse-Laboratoire d'Aérologie, Toulouse, France. The presence of insects' swarms was considered as one of the possible motivations for the problems we experienced in retrieving humidity profiles from wind profiler radar measurements. Specific references to this problem are represented by the following literature papers:
- Larkin, R. 1991. Flight speeds observed with radar, a correction: slow 'birds' are insects. Behav. Ecol. Sociobiol. 29: 221–224.
- Chapman, J.A., Reynolds, D.R. & Smith, A.D. 2003. Vertical-looking radar: a new tool for monitoring high-altitude insect migration. Bioscience 53: 503–511.
- Chapman, J.W., Drake, V.A. & Reynolds, D.R. 2011. Recent insights from radar studies of insect flight. Ann. Rev. Entomol. 56: 337–356.
- Gauthreaux, S.A., Jr, Livingston, J.W. & Belser, C.G. 2008. Detection and discrimination of fauna in the aerosphere using Doppler weather surveillance radar. Integr. Comp. Biol. 48: 12–23.

The above references have now been introduced in the text and the corresponding sentence has been partially revised as follows: "Additionally, ABLH estimates from wind profilers are sensitive and occasionally affected by the presence of insects' swarms (Larkin, R. 1991; Chapman et al., 2003; Chapman et al., 2011; Gauthreaux et al., 2008)."

Conversely, the accuracy of Doppler radar wind retrievals can also be assessed using insects as targets (among others, Rennie, S.J., Illingworth, A.J., Dance, S.L. and Ballard, S.P. (2010), The accuracy of Doppler radar wind retrievals using insects as targets. Met. Apps, 17: 419-432. https://doi.org/10.1002/met.174).

- Identification of fluctuation in wind: why is RS the only accurate method ?

The referee is right to highlight that the RS is not the only accurate method for the identification of wind fluctuation. Wind fluctuations can also be identified in wind lidar data, in wind profiler data and, for example, in in-situ sensors measurements from tethered balloons or sensors on-board scientific aircrafts. The corresponding sentence has been changed as follows: "Such fluctuations can be identified in wind lidar and wind profiler data, but measurements can also be performed with radiosondes and tethered balloons or in-situ sensors on-board scientific aircrafts."

- Line 84: The stable layer at the top of the mixed layer stops the turbulent eddies from further rising. Is it the right answer ?

The sentence is former line 84, together with a large portion of the paragraph were it was embedded, has now been removed from the text.

- Line 87-88: "Additionally, radiosonding data can provide a long observational record, which is particularly suited for ABLH climatological studies (Madonna et al. 2021)." This does not matter for this paper since only one month of observation is used.

The present sentence, together with a large portion of the paragraph were it was embedded, has now been removed from the text.

- Line 92-93: so called "bulk Richardson number for the entire ABL": the bulk Richardson number is well defined. I do not see why it is called here "bulk Richardson number for the entire ABL" ? Why to add " for the entire ABL"?

We agree that it was not correct to specify " for the entire ABL", so this portion of sentence has been removed. Now the sentence reads: "This method assumes the ABLH to be the level where the so called "bulk Richardson number" exceeds a specific threshold value, $Ri_{bc}$."

- Line 95-96 "Such gradients can be revealed in wind lidar, wind profiler, radiosonde and aircraft in- situ sensors' profile data (Sicard et al., 2006)." This sentence is not completely right. First, the bulk Richardson cannot really be considered as a gradient. Second the wind lidar and the wind profiler does not suit, alone, to the Rib calculation, that needs temperature and wind compounds. Radio-sounding does not need further wind measurement (wind profiler/lidar) since it usually also measures wind.

This incorrect sentence has now been removed from the text.

- Line 97-98: the low-level jet cannot be considered as an ABLH.

This sentence, together with a large portion of the paragraph were it was embedded, has been removed from the text. In fact, the description of the wind shear profile approach to estimate the ABLH, which relies on measurements of the vertical wind profile, in the revised version of the paper is no longer present.

- In the introduction, it is not mentioned that the raman lidar BASIL also measured temperature profiles.

It is indeed true that the temperature measurement capability of the Raman lidar BASIL had not been mentioned in the Introduction of the paper. This gap has now been filled. The following sentence has now been introduced in the paper. "As the estimate of the Richardson number requires information on both wind and thermodynamic profiles, the application of this approach to the Raman lidar relies on thermodynamic profile measurements from this sensor and wind measurements from the simultaneous and co-located radiosondes, as the wind measurements are not available from the Raman lidar." Further down in the introduction, the following sentence has also been introduced: "The capability of the Raman lidar BASIL to perform high-resolution and accurate profile measurements of atmospheric temperature and water vapour, as well as particle backscatter profile measurements, allows to obtain ABLH estimates based on both the application of the Richardson number approach and the use of elastic backscatter lidar signals."

- Line 120: it would be nice to have detailed information on the applied vertical and temporal smoothing applied to the water vaoupe and temperature profiles.

This information has now been introduced and the text has been changed as follows: "Data are sampled with a rough vertical and temporal resolution of 7.5 m and 10 sec, respectively, but vertical and temporal smoothing is typically applied when processing water vapour, temperature and particle backscatter measurements for the purpose of estimating the ABLH. In the present study we considered a vertical and temporal resolution of are 30 m and 5 min, respectively. Based on these vertical and temporal resolutions, water vapour and temperature profile measurements from BASIL extend from the proximity of the surface (50-100 m above station level) up to ~4/~10 km (day/night) and ~6/~20 km(day/night), respectively."

- Line 127: the strongest echoes in the ABL are due to higher aerosol concentration. Aerosol is the real measured parameter, not the echoe.

The sentence has been changed as follows: "An extensively used methodology relies on the detection of vertical gradients in elastic backscatter lidar signals, associated with aerosol concentration gradients"

- Line 138-140: the sentence at line 140 is completely right, but this does not correspond exactly to equation (3). The sentence line 138 is not precise enough, even it is complete thereafter at line 140.

The sentence in line 138 has been changed as follows: "As the larger vertical variability of the RCS is associated with aerosol vertical gradients, the ABLH is estimated from the height derivative of RCS through the expression: $ABLH = min\left\{\frac{d}{dz}\left[\log(RCS(z))\right]\right\}$ (5)". The sentence in line 140 has been changed as follows: "Transitions between different aerosol layers are identified with the minima in expression (5), with the highest amplitude minimum, i.e. the largest aerosol gradient, typically indicating the ABLH."

- 2.2 the ABL layers detected by the described gradient method correspond to CBLH in the mid-day and to RL during the night. This should be better described and explained. This comment complement the first main comment.

This aspect has now been properly stressed in the paper, where the following sentences have been introduced: "This approach relies on the strong sensitivity of elastic backscatter lidars to suspended aerosol particles and their gradient and on the property of aerosols to act as tracers of atmospheric motions. It is to be specified that ABLH estimate determined through this approach identifies the convective boundary layer height during the day, when convective activity is on, and the residual layer during the early morning, the late afternoon and the night, when convective activity is strongly reduced or suppressed."

- Line 143-144: why the wind profiler impeded the detection of shallow ABLH and no the lidar overlap effects ? Please be more precise in your descriptions.

The ABLH estimate from the wind profiler is based on the turbulence method, with the turbulent region being determined by tracking the fluctuations of the different wind components (*U*, *V*, and *W*). Such tracking becomes ineffective within the surface atmospheric layer as a result of the lack of sensitivity in wind measurements in this region. On the other hand, overlap effects in elastic backscatter lidar measurements may determine a compression of the elastic signal dynamics, and consequently a reduction of the amplitude of the detected range-corrected signal gradients, but will not cancel these gradients, making the detection of the ABLH still effective. This aspect is now more clearly specified in the text, where the following sentences have been introduced: "In fact, overlap effects may determine a compression of the elastic signal dynamics, and consequently a reduction of the amplitude of the detected range-corrected signal gradients, but will not cancel these gradients, making the detection of the ABLH still effective. Overlap effects are more pronounced when determining the night-time ABLH, especially when this height is within a few tens of meters and may fall in the lidar blind region."

- Line 170-172: Is it possible to have an uncertainty estimation on the ABLH from the uncertainty in the temperature profile?

This sentence, together with a large portion of the paragraph were it was embedded, was dedicated to the description of the approach to estimate the ABLH based on the application of the temperature gradient method, which relies on the identification of maxima in the potential temperature vertical gradient. The illustration of this approach has been removed from the paper because it is no longer used in its present version.

- Line 181 "agin"?

The sentence where this typing error was present has been removed from the paper.

- Line 199-201: the reader does no more know, which method is used in EAR 5.

As already illustrated above, in the revised version of the paper we are no longer making use of the ABLH products directly generated by the ECMWF-ERA5 reanalysis, but we are determining the ABLH by applying the Richardson number approach directly to wind and thermodynamic profile analysis data. So, the ABLH estimate from ECMWF-ERA5 reanalysis data reported in the present paper are indeed determined based on the application of the Richardson number method, as clearly stated in the sentence in lines 199-201. However, this sentence has now been removed from the text, together with a large portion of the paragraph were it was embedded, in order to shorten, lighten and increase the readability of the text.

- Line 201-204: it the authors want to speak about the uncertainties of the algorithm used in ERA5 (the parcel method as given at line 198), the considered uncertainties have to be described. The examples given at lines 203-204 are, to my knowledge, never considered in the parcel method.

This sentence, together the paragraph were it was embedded, has now been removed from the text. As already mentioned above, we are now are no longer making use of the ABLH products generated by the ECMWF-ERA5 reanalysis, but we are determining the ABLH by applying the Richardson number approach directly to wind and thermodynamic profile analysis data.

- Line 205: finally, the bR method is used ?? then what is the usefulness of lines 198-2004 ?

The reviewer is right: in the previous version of the paper there was some useless text related to the description of the standard ABLH products generated by the ECMWF-ERA5 reanalysis. As already mentioned above, this paragraph has been now removed as in fact the ABLH estimate presently reported in the paper is obtained from the ERA5 reanalysis by applying the Richardson number approach directly to wind and thermodynamic profile analysis data.

- Line 208-2010: it seems that the authors have not really understood the method. Rib is compute for all heights and not only for the ABLH. ABLH is given when Rib= the chosen threshold. This comment is another complement to the first main comment. It really seems that the authors have a too low knowledge of the ABL structure and the methods used to measure it.

This paragraph has been completely rewritten and moved into a dedicated sub-section (2.1 Richardson number method) within a new section (2 Methods considered for the determination of the ABLH) dedicated to methods used throughout the paper to determine the ABLH. This sub-section has been reformulated as follows:

[revised manuscript text omitted]

All these references have now been introduced in the text.

- Equation 6 is not needed since it's the same as equation 5 at another level.

Former equations 5 and 6 has now been removed from the text.

- Line 222: mothers ?

This sentence has been corrected and moved to the Conclusions (section 5).

- Line 232: I really do not see the scientific meaning to average the SBLH from 09:00 to 21:00 UTC ? if you want to describe the ABL dynamic, you should describe the growth, the maximum and, if the method allows it, the decrease of the ABLH. To compare mean over the complete convective diurnal cycle does not bring any reliable information.

As already specified above, and we really apologize for that, there had been a miscommunication among the authors. The author who finally reviewed the paper (Paolo Di Girolamo) had not properly interpreted information coming from the other coauthors taking care of the data analysis. In fact, the ABLH is not estimated as a mean over the entire day, but a much shorter time interval (half hour) is considered. The considered approach was to concentrate on the specific times when the RS data were available, typically at 00:00 and 12:00 UTC. So, all considered sensors and models are typically averaged over a half hour time interval centered on the comparison time (00:00 UTC and 12:00 UTC), i.e. 23.45-00:15 UTC for night-time comparisons and 11:45-12:15 UTC for daytime comparisons.

- Moreover, the gradient method applied to the Raman Lidar aerosol range corrected backscattering will monitor the residual layer during part of the morning and in the late afternoon, allowing no comparison with the other methods.

This aspect has been now clearly specified in the text, where the following sentence has been introduced: "It is to be specified that ABLH estimate determined through this approach identifies the convective boundary layer height during the day, when convective activity is on, and the residual layer during the early morning, the late afternoon and the night, when convective activity is strongly reduced or suppressed."

- Figure 2a: the y labeling should be ABLH and not only altitude

The y labeling has been corrected.

- Figure 2b and c, Fig. 3: why to choose the mean of all method as a reference ? usually the most reliable method (often the radio-sounding) is taken as the reference.

As already specified above, following the suggestion of the referee, we are now using the Richardson number approach applied to the on-site radiosonde data as reference.

- Line 241: Up to now the authors claimed to analyze the month of October. Why August is now mentioned? There cannot be any influence of August weather conditions on the ABLH of October.

This misprint has been removed, together with all the paragraph where it was embedded. The paragraph had been originally introduced to motivate the miss-agreement between ABLH estimates from ECMWF-ERA5 reanalysis data and all other ABLH estimates, miss-agreement that – as specified in detail above - is no longer present.

- Lines 252-254: is the EAR5 removed for the whole period or only for the second half of October

As already specified above, in the revised version of the paper we are no longer making use of the ABLH products generated by the ECMWF-ERA5 reanalysis, but we are determining the ABLH by applying the Richardson number approach directly to wind and thermodynamic profile analysis data. The present use of the reanalysis data leads to ABLH estimates which are in much higher agreement with the other five ABLH estimates. As a result of this modification, all ABLH estimates from ECMWF-ERA5 reanalysis data are used and none of them has been removed. Furthermore, the mean ABLH of all methods is no longer used as reference, but the ABLH estimate obtained from the application of the Richardson number approach to the on-site radiosonde profiles is used as reference, so also this potential source of bias has been removed.

- Line 263: if ERA5 is the only method removed from the mean to make the correlation, it is obvious that it will obtained the lowest $R^2$. (new value are present now)

This sentence has been removed for the motivations already illustrated in the previous point.

- Lines 265-275: this is not the right way to present results in an attractive way.

This paragraph has been completely re-written and now results are presented in a more attractive way, underlining the qualitative value of those results that had been previously reported only in a quantitative way. In addition, this paragraph has been expanded with the introduction of the comparisons between daytime and night-time performance for all sensors/models/approaches.

- Line 306-307: All remote sensing observation figures are always constructed like that. There is no use to describe this in the text and in the figure caption.

This sentence has been now modified both in the text and removed in the figure caption.

**Referee #2**

1) I recommend that you add some analysis of the results so they are not only quantitative but also qualitative. What does it mean that the ABLH estimates are similar? What does this say about

thermal, kinematic and aerosol definitions of ABLH? What do the results say about the model ability to measure ABLH? Is there some time of day that the results differ? It might be interesting to show an hourly composite comparison. Are there some specific weather conditions that the results differ?

We went through a complete rewriting and reshuffling of a large portion of the paper. As a results of this, results are now presented in a more attractive way, underlining the qualitative value of the results that had been previously reported only in a quantitative way. Aspects related to the similarity of the ABLH estimates and the significance of the different (thermal, kinematic and aerosol definitions) are now carefully stressed in the paper and an assessment of the model ability to simulate the ABLH and its evolution is also provided. We have now extended the inter-comparison to both daytime and night-time data, with results revealing slightly larger biases affecting daytime comparisons with respect to night-time. The use on an extended time case study covering two daily cycles and different weather conditions allow to assess the performance in monitoring the short-term variability of the ABLH. See more details below and in the revised text.

2) Why do you use the mean of the ABLH estimates as the reference ? This may need some more justification, especially since the ERA5 method has such poor results.

The approach of estimating the bias with respect to the mean value of all sensors and approaches is valid only in case a higher reliability cannot be attributed to any of the available sensors/approaches. However,  the Richardson number approach applied to the on-site radiosonde measurements is probably the most reliable approach among those considered in the present research effort, with the lowest bias, and so we can consider this approach as bias-free and take it as reference when assessing the bias of all other ABLH estimates. Consequently, in the revised version of the paper that we have just submitted, the mean of the ABLH estimates is no longer used as reference.
Additionally, in the revised version of the paper we are no longer making use of the ABLH products directly generated by the ECMWF-ERA5 reanalysis, but we are determining the ABLH by applying the Richardson number approach directly to the wind and thermodynamic profile analysis data. The present use of the reanalysis data leads to ABLH estimates which are in much higher agreement with the other five ABLH estimates. As a result of this modification, all ABLH estimates from ECMWF-ERA5 reanalysis data are used and none of them has been removed.

3) Why do you average results over a 12 hour period ? Perhaps hourly averages would be more interesting so that you could compare the measurement techniques during the boundary layer evolution.

Here there was, and we really apologize for that, a miscommunication among the authors. The author who finally reviewed the paper (Paolo Di Girolamo) had not properly interpreted information coming from the other coauthors taking care of the data analysis. In fact, the ABLH is not estimated as a mean over the entire day, but a much shorter time interval (half hour) is considered. The considered approach was to concentrate on the specific times when the RS data were available, typically at 00:00 and 12:00 UTC. So, all considered sensors and models are typically averaged over a half hour time interval centered on the comparison time (00:00 UTC and 12:00 UTC), i.e. 23.45-00:15 UTC for night-time comparisons and 11:45-12:15 UTC for daytime comparisons.

**Specific comment:**

Lines 45-48 : This is a confusing description of ABLH.

These confusing sentences have now been removed.

Lines 93-96 : This is unclear to me : The Richardson number gradient method requires observations of profiles wind speed and potential temperature. Perhaps you could clearly state that it can be calculated using radiosondes but only at limited times during launches. And it can be calculated using in situ aircraft profiles but only during ascent and descent. The wind lidar and wind profile continuous observations do not provide the potential temperature information. Please be more clear in that discussion.

These aspects are now more clearly specified in the text. In the revised version of the paper we are now comparing six ABLH estimates, i.e. the ABLH estimates obtained based on the application of the Richardson number technique to Raman lidar and radiosonde measurements and to ECMWF-ERA5 reanalysis data to the ABLH measurements from the wind profiler, which rely on the turbulence method, and from elastic backscatter lidar signals.
The Richardson number can be calculated from the wind speed and the potential virtual temperature profiles. The vertical profile of the potential virtual temperature can be determined from the vertical profiles of atmospheric temperature, pressure and water vapour mixing ratio. These profiles are available from all sensors/models involved in the inter-comparison effort, namely the Raman lidar, the radiosondes launched on-site and those and from the closest IGRA radiosonde station and the ECMWF-ERA5 analysis data, with the only exception of the wind profiler. The vertical profiles of the horizontal wind-speed components $u/(z)$ and $v(z)$, which are needed to quantify the bulk Richardson number are also available from the same sensors/models, with the only exception of the Raman lidar, which only provides the humidity and temperature profile measurements needed to determine the virtual potential temperature profile. In this case, the computation of the Richardson number is completed with the inclusion of wind-speed profile measurements from the simultaneous and co-located radiosondes launched in Candillargues.

Lines 97-99 : Why do you describe the ABLH using the LLJ technique ? Do you use this technique ? Is that valid only at nighttime ? You do not show any nighttime ABLH measurements in your comparisons. Perhaps you mention this information for completeness ? If so, please state that.

These sentence, together with a large portion of the paragraph were it was embedded, has been removed from the text. In fact, the description of the wind shear profile approach to estimate the ABLH, which relies on measurements of the vertical wind profile, is no longer present in the revised version of the paper. Additionally, in the revised version of the paper the inter-comparison is extended to both daytime and night-time cases, with slightly larger biases observed for daytime comparisons with respect to night-time.

Line 121 : What is the minimum and maximum height range for BASIL ?

The minimum and maximum height range for BASIL may vary in dependence of the application. For example, measurements of the vertical profiles of atmospheric temperature and water vapour mixing ratio, which are based on the application of the rotational and vibrational Raman lidar techniques, respectively, are obtained by ratio-ing two signals, i.e. the low- and high-quantum number rotational Raman signals from molecular nitrogen and oxygen in the case of temperature measurements and the water vapour and the molecular nitrogen roto-vibrational Raman signals in the case of water vapour mixing ratio measurements. As the ratio-ed signals have very similar overlap functions (and this is guarantee by the very compact optical design of the system), overlap effects tend to cancel out and temperature measurements and the water vapour may extend down to the blind region (50-100 m above station level).

The text has been integrated as follows: "Data are sampled with a rough vertical and temporal resolution of 7.5 m and 10 sec, respectively, but vertical and temporal smoothing is typically applied when processing water vapour, temperature and particle backscatter measurements for the purpose of estimating the ABLH. In the present study we considered a vertical and temporal resolution of are 30 m and 5 min, respectively. Based on these vertical and temporal resolutions, water vapour and temperature profile measurements from BASIL extend from the proximity of the surface (50-100 m above station level) up to ~4/~10 km (day/night) and ~6/~20 km(day/night), respectively."

For what concerns the ABLH estimate from elastic backscatter lidar signals, it is to be specified that overlap effects in elastic backscatter lidar measurements may determine a compression of the elastic signal dynamics, and consequently a reduction of the amplitude of the detected range-corrected signal gradients, but will not cancel these gradients, making the detection of the ABLH still effective.

This aspect is now extensively addressed in the text and the following sentences have been introduced: "Overlap effects affect lidar signals in the lower few hundred meters, but have marginal effects on gradient measurements and consequently the accuracy of ABLH estimates. In fact, overlap effects may determine a compression of the elastic signal dynamics, and consequently a reduction of the amplitude of the detected range-corrected signal gradients, but will not cancel these gradients, making the detection of the ABLH still effective. Overlap effects are more pronounced when determining the night-time ABLH, especially when this height is within a few tens of meters and may fall in the lidar blind region."

Line 143 : If there are no data in the lower few hundred meters, could that limit your observations of nocturnal ABLH ? Perhaps you could state here that you are not measuring nocturnal ABLH ?

As already specified in the previous point, this aspect has now been extensively addressed in the text and the following sentences have been introduced: "Overlap effects affect lidar signals in the lower few hundred meters, but have marginal effects on gradient measurements and consequently the accuracy of ABLH estimates. In fact, overlap effects may determine a compression of the elastic signal dynamics, and consequently a reduction of the amplitude of the detected range-corrected signal gradients, but will not cancel these gradients, making the detection of the ABLH still effective. Overlap effects are more pronounced when determining the night-time ABLH, especially when this height is within a few tens of meters and may fall in the lidar blind region."

Line 157-159 : You note that the applicability of the ABLH technique from a wind profiler can be limited by strong reflectivity peaks due to temperature and humidity gradients. I must be misunderstanding the wind profiler technique because I thought the gradients are what determines the reflectivity peaks. Please further explain the wind profiler technique and how the reflectivity peaks are different from the temperature and humidity gradients. Also please describe how « This aspect will be carefully accounted for .. . »

We apologize for the misleading content and inaccurate writing of these sentences, which have now been removed. As suggested by the referee, we have now also further explained the wind profiler technique and how the reflectivity depend on atmospheric thermodynamic properties. The corresponding sentences have been changes as follows: "The methodology applied to determine the ABLH relies on the identification of a distinctive strong peak in the WPR time-height reflectivity plot (Gage et al., 1990), which is associated with turbulence-generated radio refractive index fluctuations, associated atmospheric thermodynamic parameters' fluctuations, though the strength of this peak may depend also on other factors. Wind profiling radars are sensitive to scales of turbulence that equal half the radar wavelength. At 1.274 GHz, this wavelength is ~20 cm. Therefore the wind profiler is sensitive to turbulent eddies with spatial dimensions of ~10 cm

causing fluctuations in the radio refractive index. In the boundary layer, essentially all scales of turbulence exist from 1-cm wavelengths up. Wind velocity measurements rely on the detection of the Doppler shift, assuming that fluctuations in the radio refractive index, dependent on atmospheric thermodynamic parameters, are carried along the mean wind flow."

Figure 5a : Could you please add the results of the other methods of determining ABLH to this figure ? Why does the red line appear to be significantly lower than the maximum aerosol gradient ?

All six ABLH estimates considered in the paper have now been introduced in both figures 5a and 5b.

Figure 5b : Could you please also add the ABLH results on Fig 5b ?

All six ABLH estimates considered in the paper have now been introduced in both figures 5a and 5b.

**Technical and typographical comments:**

Line 52 : Please change to « . . . of recent technological progress. . . . »

This portion of sentence has now been removed from the text.

Line 53 : Please insert « such » into here : « . . .atmospheric variables, such as particle. . . . »

Modified in the way suggested by the referee.

Line 144 : Please add 'and' here « . . . meters and have marginal . . . »

Corrected

Line 144 : Please quantify « marginal effects ».

This aspect is now properly addressed in the text, where the following sentences have been introduced: "In fact, overlap effects may determine a compression of the elastic signal dynamics, and consequently a reduction of the amplitude of the detected range-corrected signal gradients, but will not cancel these gradients, making the detection of the ABLH still effective. Overlap effects are more pronounced when determining the night-time ABLH, especially when this height is within a few tens of meters and may fall in the lidar blind region."

Line 173 : Please delete « again ».

This sentence, together with the paragraph where it was embedded, has been removed from the text. This sentence referred to operation details in the application of the temperature gradient method, which is no longer considered in the present research effort among the approaches applied to the measurements/model data.

Line 173 : Also, what are marginal effects ? Please be more quantitative.

This sentence, together with the paragraph where it was embedded, has been removed from the text. This sentence referred to operation details in the application of the temperature gradient method,

which is no longer considered in the present research effort among the approaches applied to the measurements/model data.

Line 181 : Suggested modification : « . . .radiosondes are obtained by using the temperature gradient method. »

This sentence, together with part of the paragraph where it was embedded, has been removed from the text. This sentence referred to the application of the temperature gradient method, which is no longer considered in the present research effort among the approaches applied to the measurements/model data.

Line 185 : I recommend that you delete « again »

This portion of sentence has been removed. It was referring to the application of the temperature gradient method, which is no longer considered in the present research effort among the approaches applied to the measurements/model data.

Line 185 : Please quantify what you mean by « negligible uncertainties »

This portion of sentence has been removed. It was referring to the application of the temperature gradient method, which is no longer considered in the present research effort among the approaches applied to the measurements/model data.

Line 209 : Please correct the typo at the end of the line and change « here » to « where »

Corrected in the sentence, but the text of this paragraph has been partially reshuffled.

Line 216 : I recommend that you delete « being the »

Corrected in the sentence, but the text of this paragraph has been partially reshuffled.

Line 222 : Please delete « the » between « observed » and « in »

This sentence has been removed from the text, together with part of the paragraph where it was embedded.

Line 222 : What is meant by « and sensors and mothers » ?

This sentence has been corrected and moved to the Conclusions (section 5).

Line 232 : Please tell us the LT for 0900 – 2100 UTC.

As already specified above, the time window from 9:00 to 21:00 UTC was an erroneously reported information. In fact, the ABLH is not estimated as a mean over the entire day, but a much shorter time interval (half hour) is considered. The considered approach was to concentrate on the specific times when the RS data were available, typically at 00:00 and 12:00 UTC. We are now considering a separate comparison for daytime and night-time cases, at 12:00 UTC and 00:00 UTC, respectively. Local times for these UTCs have now been specified in the text and the corresponding sentence reads as follows: "A separate comparison has been carried out for daytime and night-time cases, at 12:00 UTC and 00:00 UTC, respectively (local time is UTC+02:00 hours in this period of the year)."

Line 234 : Please correct the typo in the word « its » here : « activity, from its activation . . . «

The present sentence has been removed from the text.

Line 234 : I recommend that you remove the words « a quite ».

Corrected.

Line 257 : I suggest you change it to « scatter plots ».

Corrected.

Line 286 : Please change to « . . .50-100 ».

Corrected.

Line 287 : I think that « unfavorable » should be one word.

Corrected.

Line 303 : Please modify to : « . . .corresponding relative humidity values. »

Corrected.

Line 306 : I suggest you change the word « map » to « figure ».

Corrected.

Line 332 : Please remove the word « the » before « them. »

Corrected.

Line 334 : Please correct the typos : « . . .computed from different sensors are in the range. . . »

Corrected.

Line 343 : There are two periods at the end of the sentence.

Corrected.

Figures 2b and 2c : It would be helpful to add a horizontal line at bias=0.

A black dashed horizontal line has now been introduced in all four figure panels illustrating bias values (panels $b_1$ and $c_1$ for daytime comparisons and panels $b_1$ and $c_1$ for night-time comparisons).

Line 487 : Suggested modification : « . . .expreseed in terms of scatter plots, . . . »

Corrected.

Figure 3 caption : It looks like the font in the lines 487-488 is different from the font in the rest of the manuscript.

Corrected

Figure 3b : Where is the best fit line in Fig. 3b ?

In the previous version of figure 3, we had introduced in all panels the 1:1 bisector line for the purpose of highlighting the deviation between the fitting lines and those corresponding to zero bias. In the case of figure 3 b the agreement was so good that the fitting line was hidden by the 1:1 bisector line. In order to avoid such hiding, 1:1 bisector lines have now been removed from these figure panels.

Figure 5 : Please add (a) and (b) to the figure panels.

The indication (a) and (b) has been added to the figure panels.

Figure 5b : The x-axis is missing the first « 9 ».

The time labels in figure 5b have been corrected.

Figure 5 caption suggestion : « Figure 5 : (a) Time-height . . . ., (b) and water vapour mixing ration over the time . . . »

Corrected following the suggestions of the referee.

---

## Author Response (AR2)

Dear Editor,

We are very grateful to referee #1 and the Associate Editor for their additional constructive suggestions and for their proposed corrections. We have addressed all issues raised and have modified the paper accordingly. We have also submitted a revised version of the paper where all these changes have been incorporated. We believe that, thanks to these precious inputs, the quality of the manuscript has sensitively improved. Below is a summary of the changes we carried out and our specific responses to the referee' comments and recommendations.

**Summary of the changes**
**(in black is the original comments of the referee and in red our responses)**

« Inter-comparison of ABL height estimates from different profiling sensors and models in the framework of HyMeX-SOP1" by Summa et al.
The manuscript greatly improved and I do appreciate the large effort of the authors to improve the content, the structure, the figures and the language of the paper. To my opinion, this second version reaches the quality of a first submitted version. There is however still a lot of scientific inaccuracies

The scientific inaccuracies to which the reviewer is referring to have been removed also based on the specific suggestions from his side in this direction.

such as e.g. error in the sign of the mean bias,

We removed this inaccuracy based on the specific suggestion from the reviewer. Specifically, the following text was corrected/introduced: "More specifically, very smallest biases are found to characterize the ABLH estimates obtained through the application of the Richardson number method (18.2 m /7.4 % for the IGRA radiosondes, 20.5 m/8.15 % for the Raman lidar BASIL, and -28 m/-2.97 % for ECMWF-ERA5 analysis), while slightly large bias values are found to characterize ABLH estimates from the wind profiler (47 m/21.7 %) and ABLH estimates from elastic backscatter lidar signals (61.6 m/26.4 %). The slightly smaller ABLH values characterizing ECMWF-ERA5 analyses are probably to be attributed to the systematically smaller values of the water vapour mixing ratio and the wind V component from ECMWF-ERA5 with respect to those from other sensors. Another possible motivation for the negative systematic bias affecting ABLH estimates from ECMWF-ERA5 is represented by the missed assimilation of the on-site radiosondes in ECMWF-ERA5." See below more specific comments/replies with reference to specific "minor comment" from the reviewer in L258.

designation of the weather as "clear-sky" when clouds are presents,

We removed this inaccuracy based on the specific suggestion from the reviewer in this direction Specifically, the corresponding sentence was modified as follows: "The analysis was also focused on one specific case study, covering an extended time interval from 09:00 on 18 October 2012 to 19:00 UTC on 19 October 2012, including two daytime portions, the first one

characterized by the presence of high scattered clouds between 3 and 4 km and the second one characterized by the present of low stratiform clouds between 1 and 2 km, which allowed to assess the performance in the characterization of the short-term variability of the ABLH in variable weather conditions." See below more specific comments/replies with reference to specific "minor comment" from the reviewer in L395.

the description of a pressure measurement by the Raman Lidar

We removed this inaccuracy based on the specific suggestion from the reviewer in this direction. Specifically, the following sentences were modified/integrated as follows: "Vertical profiles of $T(z)$, and $\chi_{H2O}(z)$ are available from all sensors/models involved in the inter-comparison effort, namely the Raman lidar, the radiosondes launched on-site and those from the closest IGRA radiosonde station and the ECMWF-ERA5 analysis data, with the only exception of the wind profiler. For most sensors, vertical profiles of $P(z)$ are obtained by vertically extrapolating the surface pressure value $P_0$.". See below more specific comments/replies with reference to specific "minor comment" from the reviewer in L109-111.

or the attribution of potential WPR problem to insects by citing studies on radar with much lower wavelengths.

We removed this inaccuracy as in fact the corresponding erroneous sentence in the paper has been deleted. See below more specific comments/replies with reference to specific "minor comment" from the reviewer in L183-185).

The first main problem mentioned (see below) concerns the description of Figures 5 and is directly linked to the first main comment of the first review: "The notion of ABL height is used in the whole introduction and attributed to several "heights" measured by various instruments and methods. The authors should really attribute the right ABL (sub)structure to the right layer height detection."

The main result of the present paper is represented by the demonstrated capability to use different sensors and models to measure the ABLH over extensive periods of time. These sensors and models were applied to an entire mouth period (the month of October 2012). In the paper, detail analysis, supported by a comprehensive statistical analysis, is carried out with the aim to properly assess the performance of different sensors and models used to estimate the ABLH over an extensive period. Illustrated results and the quality of the inter-comparison demonstrate the reliability of the combined use of these approaches over extended periods of time. In this regard, the specific case study on 18-19 October 2012 was aimed to provide a preliminary assessment of the applicability of these approaches in quantifying the short-term variability of the ABLH in variable weather conditions. So this case study is to be considered as a test-bed to assess the accuracy of these approaches, which are usually considered for a more operation use over extended measurement periods, when applied to monitor short-term ABLH variability in complex weather conditions. This analysis does not intend to be neither comprehensive nor

definitive, but to represent only a first step forward in the direction of estimating the ABLH short-term variability in variable weather conditions.

However most of the criticism of the reviewer focuses not on the main part of the paper (sub-section 4.1 "Climatological variability throughout October 2012"), by only on this more ancillary part of the paper, i.e. the analysis of the short-term variability in the period on 18-19 October 2012. Authors are available to remove this portion of the paper, if the reviewer is not happy with it, but it should be underlined that the case study on 18-19 October 2012 was only considered with the aim to illustrate the performance of operation approaches to a highly variable weather situation and was not aimed at solving all possible problems related to the short-term variability of the aerosol, cloud, wind, temperature and humidity fields. Furthermore, the operational use of the discussed approaches is in no extent compromised by the failure of their application in the monitoring of the ABLH short-term variability in variable weather conditions. This aspect is now explicitly mentioned in the conclusions, where the following sentence has been introduced: "This final analysis, while providing preliminary encouraging results, has in no extend to be considered as a thorough demonstration of the applicability of the considered approaches to variable complex weather conditions as in fact a more comprehensive and extensive study has to be carried out in the future in this direction."

The authors clearly describe now the methods and the instruments, but the ABL structure in clear-sky, cloudy conditions as well as with precipitation cannot rely only on the two quite general § of the introduction.

The introduction has been extended in the direction to properly include the different ABL weather scenarios indicated by the reviewer and illustrate the potential of the different approaches and strategies used to characterize the ABLH variability in these cases. For this purpose, specific new references have also been cited in the Introduction and the following new paragraph has been introduced in the Introduction:

[revised manuscript text omitted]

Main comments:

- The situation measured during the 18-19 October is a very complex one regarding ABLH. The authors described carefully the meteorological conditions with high clouds, precipitation and virga topped by a drier layer on the 18th, a change in wind direction on the 19th in the morning, a stratiform cloud cover with shallow orographic precipitation during the second day. The authors have to refer to scientific publication to explain what is the expected behavior of the ABL under these complex conditions.

More information is now provided concerning the expected behavior of the ABL under complex weather conditions. This is done also making use and referring to scientific publications on this topic (Che, and Zhao, 2021; Dai et al., 2014; Dang et al., 2019a; Dang et al., 2019b; Herrera-Mejía, and Hoyos, 2019; Liu et al., 2022; Manninen et al., 2018; Staudt, 2006).

Additionally, as already specified above, the introduction has been reinforced based on the inclusion of a number of new citations properly addressing the topic of the applicability of different instruments/methods/models in different meteorological and environmental conditions. The following new paragraph has been introduced:

"However, accurate estimates of the ABL height and structure in complex terrains or under complex meteorological conditions remain problematic. A variety of authors have tried to address this challenging issue. Among others, Herrera-Mejía and  Hoyos (2019) studied the spatio-temporal evolution of the ABL in a narrow, highly complex terrain located in the Colombian Andes, where convective activity, as a result of aerosol dispersion, increases the uncertainty affecting the estimate of ABLH. Staudt (2006) provided a comprehensive analysis of the ABLH variability over complex terrains in the Bavarian Alpine foreland, based on the use of multi-sensor data collected during the field experiment SALSA 2005. Che and Zhao (2021) assessed the effectiveness of different approaches to characterize the summer ABLH variability over the Tibetan Plateau, this region being characterized by elevations exceeding 4000 m and complex land surface processes and boundary layer structures.
Coming to the characterization of the ABLH in cloudy conditions, Dang et al. (2019) investigated different approaches, with a specific focus on reducing the interference of the residual and cloud layers on ABLH determination. Manninen et al. (2019) demonstrated the capability of Doppler lidars to determine the ABLH in both clear-sky and cloud-topped conditions, with some reservations in precipitation. Furthermore, Liu et al. (2022) proposed an

approach to estimate ABLH from elastic backscatter lidar data under complex atmospheric conditions based on the use of machine learning methods.

However, both the results from these previous papers and the conclusions reached in the present paper clearly indicate that a proper characterization of the ABL height and structure in all weather conditions requires the combined application of different approaches and data sets. This approach allows to overcome the possible dependence of each single sensor/method from a specific meteorological parameter, thus drastically reducing potential biases affecting ABLH estimates (Dai et al., 2014b). Additionally, multi-sensor approaches have demonstrated to be more robust and better performing in variable stable and unstable weather conditions (Joffre et al. 2001)."

For example, the ABLH increase at around 00:00 does it correspond to a real change in atmospheric boundary layer height or to a change in advected humidity/aerosol due to wind direction change ?

We are now we are now properly interpreting the ABLH increase observed around 00:00 UTC. This is associated to a real change in the ABLH caused by advection of air masses with different thermodynamic and compositional properties, associated with a wind direction change and altering the stability and turbulent state of the atmosphere. Similar considerations were reported by Pal and Lee (2019), who suggested that contrasting air mass advection associated with onshore and offshore flows may explain the significant variability in ABLH.

This aspect is now more extensively illustrated in the paper, where the following sentence was introduced: "This change in the ABL structure is caused by advection of air masses with different thermodynamic and compositional properties associated with the wind direction turning, ultimately altering the stability and turbulent state of the sounded air masses. A similar abrupt ABLH variability had been reported by Pal and Lee (2019), who revealed the important role played in coastal areas by advection via onshore and offshore flows."

Pal, Sandip & Lee, Temple. (2019). Contrasting Air Mass Advection Explains Significant Differences in Boundary Layer Depth Seasonal Cycles Under Onshore Versus Offshore Flows. Geophysical Research Letters. 46. 10.1029/2018GL081699.

It is also important to know the stability conditions and how do the precipitation/clouds affect the various measurements/detection methods?

Stability conditions have been carefully investigated, based on observations and model data, which indicate that a higher correlation is present in case of stable weather conditions. An additional graph (new figure 5) has been added that distinguishes stable days from unstable ones. Specifically, figure 5 illustrates the percentage bias of the different ABLH estimates with respect to the one obtained through the application of the Richardson number approach to the on-site radiosonde profiles, expressed as a function of the atmospheric stability conditions. More than 50 % of the cases can be classified as stable condition, while the reminder cases can be classified as

unstable. For the unstable conditions mutual deviations between the different/sensors methods are approximately 30% larger than in the case of stable conditions. A lower dispersion of the bias values implies a higher correlation among the different ABLH estimates in case of stable weather conditions.

Intense measurements during these days allows having temperature, humidity, wind and aerosol backscatter profiles. I'm e.g. very interested in an analysis allowing to know which of the used T, RH and wind profiles leads to the rapid change in ABLH at 00:00 detected by the bulk Richardson method?

We have verified that in this specific case the variability of temperature has limited effects on the calculation of $R_{ib}$, while large are the effects of the wind V component, varying significantly starting at 23.30 on 18 October. This translates into with friction velocity abruptly rising starting at 23.30. Below follows a plot of friction velocity, where this abrupt increase is illustrated.

[Figure]

We also computed the mean deviations between the ERA5 and the on-site radiosondes for water vapour mixing ratio, temperature and the wind V component throughout the month of October 2012. A 50 % mean deviation, with ERA5 underestimating the on-site radiosondes, is found for the water vapour mixing ratio; a 20 % mean deviation, with ERA5 again underestimating the on-site radiosondes, is found for the wind V component, while deviations in terms of temperature are negligible. The above mentioned negative biases translate into a systematic underestimation of the coefficient $R_{ib}$, which determines a systematic underestimation of the ABLH when using ECMWF-ER5 data. This aspect is now properly illustrated in the paper, where the following sentences have been introduced: "The slightly smaller ABLH values characterizing ECMWF-ERA5 analyses are probably to be attributed to the systematically smaller values of the water vapour mixing ratio and the wind V component from ECMWF-ERA5 with respect to those from other sensors. Another possible motivation for the negative systematic bias affecting ABLH estimates from ECMWF-ERA5 is represented by the missed assimilation of the on-site radiosondes in ECMWF-ERA5."

I have also some doubts: the largest RCS gradient at 9:00 on the 18th October seems visually (Fig 5a) to be at ~1200 m and not at 800 m. Similarly I would put the RCS gradient at 00:00 at ~600 m in the continuity of the previous ABLH. Are some conditions on the size/altitude of the gradient applied?

The referee is right about these altitudes. We reviewed few specific profiles, i.e. those at 9.00 and 12.00 UTC on October 18 and at 00:00 on October 19, and we realized that the new values are closer to those indicated by the reviewer. Specifically, it can be seen that, after this last revision, the ABLH value at 9:00 becomes slightly higher, i.e. around 1140 m, while the values at 12:00 on October 18 and at 00:00 on October 19 becomes slightly lower. These new values are now included in the modified version of figure 5. For the purpose of clarity here follows the three plots of Rib at the above mentioned times.

[Figure]

[Figure]

[Figure]

- The Raman lidar is probably measuring during most of the month of October 2012. ABLH "standard" diel cycle could be then compared to the complex case of 18-19 October. The wind compound could then be taken from the WPR or the parcel method can be applied.

Authors are not sure they completely understand what the reviewer is suggesting to do here. However, ABLH "standard" diel cycle for the days 18 and 19 October, i.e. the values obtained in sub-section 4.1 for the climatological assessment throughout October 2012, are already available and introduced in figures 2 and 3 and in Table 2 (former table 1). These values reasonably well compare with those determined in figure 5 with a much higher temporal resolution. Specifically, values of the ABLH from the six sensors/models/approaches at 12:00 on 18 October 2012 and at 00:00 and 12:00 on 19 October, determined for the purpose of the climatological analysis, are 948 m, 1000 m and 1806 m, respectively, while corresponding ABLH from the six sensors/models/approaches at these same times from the short-term analysis are 750 m, 800 m, 1750 m, respectively.

- A more precise analysis of the ABLH overestimation/underestimation of the various instruments/method/model as well as the potential reasons is expected.

A new table (new Table 1) has been introduced for the purpose of providing a more comprehensive analysis of the performances of the various instruments/model/methods. More

specifically Table 1 provides PROs and CONs of the considered instruments/reanalysis and their potential sources of bias.

- Explanation of expected effect of CAPE and the relative humidity from ERA5 on ABLH should be given in order to understand why the authors chose these parameters.

We investigated the potential influence of CAPE and relative humidity on the ABLH. We are now providing supporting arguments and references to explain the effects of CAPE and relative humidity from ECMWF-ERA5 analysis on the ABLH and its variability. The following new text has now been introduced: "CAPE is strongly controlled by the ABL properties, with their coupling holding in both convective and non-convective conditions (Donner and Phillips, 2003). Additionally, the response of the boundary layer to relative humidity is investigated although it is known that it involves competing mechanisms (Ek and Mahrt, 1994), with the net effect on relative humidity being difficult to disentangle. The effect of relative humidity on the ABLH and its variability has been investigated in a variety of literature papers, with higher ABLH values typically found for high surface temperature and low humidity values, which implies surface sensible heat fluxes being dominat over latent heat fluxes and leading to increased buoyancy (Zhang et al., 2013)."

In support of the above new argument, the following new references have been introduced:

Donner, L. J., and Phillips, V. T. (2003), Boundary layer control on convective available potential energy: Implications for cumulus parameterization, J. Geophys. Res., 108, 4701, doi:10.1029/2003JD003773, D22.

Ek, M., & Mahrt, L. (1994). Daytime Evolution of Relative Humidity at the Boundary Layer Top, Monthly Weather Review, 122(12), 2709-2721. Retrieved Mar 8, 2022, https://journals.ametsoc.org/view/journals/mwre/122/12/1520-0493_1994_122_2709_deorha_2_0_co_2.xml

Zhang, Y., Seidel, D. J., & Zhang, S. (2013). Trends in Planetary Boundary Layer Height over Europe, Journal of Climate, 26(24), 10071-10076. Retrieved Mar 8, 2022, from https://journals.ametsoc.org/view/journals/clim/26/24/jcli-d-13-00108.1.xml

- The result section could be divided into some subsections

As suggested by the reviewer, we are now dividing the results section (section 4) into two separate subsections: subsection "4.1 Climatological variability throughout October 2012" and "4.2 Short-term variability over the two-day period18 and 19 October 2012".

Minor comments:

- L21:"and in the range 0.95-1.03" for nighttime comparisons? for all data ?

This aspect is now clearly specified in the text, where the corresponding sentence was modified as follows: "Values of the slope of the fitting line in the regression analysis applied to the different sensor/model pairs are in the range 0.91-1.08 for daytime comparisons and in the range 0.95-1.03 for night-time comparisons, which testifies the very small biases affecting all five ABLH estimates with respect to the reference AHBL estimate, with slightly smaller bias values found at night."

- L66: concerning the previously described methods, it is not mentioned if they also allow a detection in both daytime and nighttime. Is it the case?

This is now clearly specified in the text, where the corresponding sentence was modified as follows: " Such approach, extensively used in the present inter-comparison effort, both in day and night time, is described in detail in the following section."

- L97-98: how do you determine if the conditions are stable or convective ?

For the general application of the Richardson number method, the threshold Richardson number $R_{ibc}$ is typically taken equal to 0.25 in stable boundary layers and equal to 0.45 in unstable boundary layers. In the previous version of the paper there was a misprint as in fact we intended to mean "unstable boundary layers" and not "convective boundary layers". We considered the layers to be generally stable during the evening, night and early morning periods, while layers were assumed to be unstable during the late mornings and afternoons, in clear air conditions. These conditions were verified through the use of atmospheric stability indexes from ERA5 model analysis data. Specifically, we considered the "lifted index" LI, where negative values indicate instability - the more negative, the more unstable the air is - and positive values indicate stability.

The period September-November 2012 was selected for the field campaign because of the specific interest and focus on convection of the Hydrological cycle of the Mediterranean Experiment first Special Observing Period (HyMeX-SOP1). Within this three-month period, Intense Observation Periods (IOPs) were typically selected when convective instability conditions were present. Thus, the number of cases characterized by unstable layers is predominant in this dataset. Conditions are found to be persistently unstable during the IOP on October 18-19 and in the period October 16-22. A new figure (new figure 5) has been introduced to properly identify stable and unstable cases. These aspects are now clearly specified in the text, where the corresponding sentence was modified as follows: "We generally considered the layers to be stable during the evening, night and early morning periods, while layers were assumed to be unstable during the late mornings and afternoons in clear air conditions. The period September-November 2012 was selected for the field campaign because of the specific interest and focus on convection of the HyMeX-SOP1. Within this three-month period, Intense Observation Periods (IOPs) were typically selected when convective instability conditions were present. Thus, the number of cases characterized by unstable layers is predominant. Conditions

are found to be persistently unstable during the IOP on October 18-19 and in the period October 16-22. The presence of stable or unstable conditions was verified based on the use of a specific atmospheric stability index, i.e. the "lifted index", obtained from ERA5 model analysis data (see forthcoming figure 5))."

- L109-111: Does the Raman lidar really measure P(z) ? Most of the present radiosondes also do not measured P(z) but deduce it from the pressure at the ground and the altitude. Is it the case for your radiosonde?

In principle, pressure profiles are measured by the Raman lidar BASIL based on the combined use of the temperature profiles obtained through the rotational Raman technique and the density profiles obtained through the Rayleigh integration technique. However, we realized that the pressure profiles from the Raman lidar are often too nosy for theie effective use in estimating potential virtual temperature profile. Consequently, after having carefully checked the presence of a good agreement between the pressure profile measured by the Raman lidar and those obtained by vertically extrapolating the surface pressure value, this latter profile was used to estimate potential virtual temperature profiles to be associate to lidar measurements. This aspect is now clearly specified in the text, where the corresponding sentences were modified/integrated as follows: "Vertical profiles of $T(z)$, and $\chi_{H2O}(z)$ are available from all sensors/models involved in the inter-comparison effort, namely the Raman lidar, the radiosondes launched on-site and those from the closest IGRA radiosonde station and the ECMWF-ERA5 analysis data, with the only exception of the wind profiler. For most sensors, vertical profiles of $P(z)$ are obtained by vertically extrapolating the surface pressure value $P_0$."

- L129-133: I appreciate very much the large effort of the authors leading to a much better description of the used method. I doubt however that the elastic lidar equation is needed here. It is described in a lot of textbook. Anyhow, the authors are completely free to leave it.

We agree with the reviewer that the elastic lidar equation is described in a variety of test books and so its inclusion here is not essential. However, we decided to keep it here because, in its present formulation, it includes specific terms that are not always carefully described when discussing measurement accuracy. Among these is the term $P_{bgd}$, which represents the background signal associated with solar irradiance and detectors' noise, which is one of the major driver of the measurement uncertainty.

- L183-185: The WPR used in this study works at 1,274 GHz ( 23.3 cm). The radar described by Chapman (last publication in 2015) are radars working at much higher frequencies corresponding to wavelength between 3.2 cm and 9 mm. Gauthreaux's paper (2008) used Doppler weather radar. The weather radars usually work at frequency between 12.6 and 2 GHz (depending on the C, S or X bands) corresponding to 0.32-15 cm. Insects are detected by radars with wavelengths corresponding to the size of the insects. WPR with a wavelength of 23.3 cm

detects, to my knowledge, birds. However I will be very interested if the authors can send me a reference where insects are detected by WPR.

The present incorrect sentence has now been removed.

- L213: please add a space: radiosonde are

Corrected in the way proposed by the reviewer.

- § 3.3 and 3.4: is the vertical resolution of the launched radiosonde in Candillargues the same as the one of IGRA in Nimes-Courbessac? If not, what is the impact on ABLH?

The IGRA radiosondes are characterized by a lower resolution than  the Vaisala RS92 radiosondes. Consequently, data from the two radiosondes had to be interpolated to a common height array. The deviations associated with the interpolation procedure have been verified to be negligible. This aspect is now clearly specified in the text, where the corresponding sentence was modified as follows: "The IGRA radiosondes are characterized by a lower resolution than  the Vaisala RS92 radiosondes. Consequently, data from the two radiosondes have been interpolated on a common height array, with the deviations associated with the interpolation procedure having been verified to be negligible."

 Figures 2c1 and 2c2: please add a smaller graduation of the y-axis (and perhaps horizontal lines) so that the relative bias can be easily estimated. Figure caption should be revised by the inclusion of some abbreviations.

Both the figures layout and the caption were modified in the way suggested by the reviewer.

- L245-246: ABLH differences in daytime are comprised between -400 to +600 m (approximately -30% to +50%). ABLH differences in nighttime are comprised between -300 to +300m (~-40 to +120%). Is it a "good agreement" in all cases?

The reviewer is right: the agreement between the six different estimates cannot be considered good for all sensors/models pairs, especially when considering both daytime and night-time conditions. The term "good agreement" was modified into "general good agreement" and the corresponding sentence now reads: "Results reveal a general good agreement between the six different estimates both in daytime and night-time, all of  them revealing the major features associated with ABLH monthly variability."

More considerations are now carried out and introduced in the text. Specifically, it is to be noticed that percentage differences may be large especially during the night, when the ABL is very shallow. In our specific case, ABLH values are found to not exceed 300 m throughout most part of the month of October 2012, i.e. the period considered in our study. Nevertheless, percentage bias values in excess of 100 %, as those mentioned by the reviewer, are found only 2 times (2 data points) throughout the month of October 2012, in the comparison of the ABLH

estimates obtained from the Raman lidar BASIL with those obtained from the on-site radiosondes, considered as reference. These large values are to be attributed to the limited sensitivity of Raman lidars below 250 m, where overlap effects degrade the signals and a blind vertical region is present. These aspects are now more clearly specified in the text, where the following new sentences have been introduced: "However, percentage differences may be large especially during the night, when the ABL becomes shallow. During the month of October 2012, approximately 60 % of the night-time ABLH values are found to not exceed 300 m. Percentage differences in excess of 100 % are found only 2 times in this period and characterize the comparison of BASIL vs. the on-site radiosondes, with the former of these two sensors having a limited sensitivity below 250 m, as a result of the presence of overlap effects and a blind vertical region."

- L253-254I just count the number of Basil RCS cases with a relative error > 20% at 12:00 (Table 1 first column): 12 cases (39%). Can we consider 39% as "few data points"? I didn't count the number for the other columns.

The text has been reformulated here, with the modification of the previous sentence and the introduction of a new one: Now the text reads. "Most deviation values are within ± 200 m and ± 20 %. Again, larger deviation values are found to characterize those comparisons considering lidar-based estimates of the ABLH, as a result of the above mentioned overlap effects and the presence of a blind vertical region."

- L255: please mention "with absolute mean bias" and "relative (instead of percentage) mean bias".

The sentence has been corrected in the way proposed by the reviewer.

- L258: the mean bias are negative for ERA5 and not positive as mentioned. Please correct and try to explain why ERA5 underestimate ABLH. The radiosonding at Nice is assimilated in ERA5 ?

The reviewer is right: there was a misprint in the text as in fact values reported in Table 2 (formerly Table 1) of ERA5 vs RS at night [00:00 UTC] are negative (-28 m/-2.97 %q, but are reported in the text as positive (28 m/2.97 %). This misprint has now been corrected. Additionally, we are now trying to explain why ERA5 underestimate ABLH. The following new sentence has been introduced: "The slightly smaller ABLH values characterizing ECMWF-ERA5 analyses are probably to be attributed to the systematically smaller values of the water vapour mixing ratio and the wind V component from ECMWF-ERA5 with respect to those from other sensors. Another possible motivation for the negative systematic bias affecting ABLH estimates from ECMWF-ERA5 is represented by the missed assimilation of the on-site radiosondes in ECMWF-ERA5."

- L262: please rephrase "also at night"

This portion of sentence has been rephrased as follows: "Again, smaller biases are found to characterize night-time ABLH estimates obtained through the application of the Richardson number method … "

- L269-274: the description of the linear regression could be shortened

The description of the linear regression has been shortened by approximately 30 % and now reads: "A linear fit is applied to the data points, using a linear regression function passing through zero, with the form $Y = A \times X$. X are the ABLH reference values and Y are the values from the five reminder ABLH estimates. The term A represents the slope of the fitting line and provides an alternative estimate of the bias of each of the five ABLH estimates with respect to the reference, while the correlation coefficient $R^2$ of the linear fit quantifies the degree of agreement between the compared ABLH estimates."

- Figure 3 and Table 2: the values in Table 2 could be directly reported in Figures 3. The one-one line in all figures could also help (see first version of the manuscript).

We agree that values in Table 2 could be directly reported in the different panels of Figure 3, as it was in the first version of the manuscript. However, the request to the introduce a separate table summarizing all results from the regression analysis was coming from one of the other reviewers during the previous revision round and a removal of this table now would represent a disregard of that specific reviewer request.

For clarity reasons, please use the word "slope" instead of the A throughout the manuscript.

Now the term "slope" in used in substitution to the term "A" throughout the manuscript.

- L278: AHBL

This misprint was corrected.

- L280-295: a more dynamic description of the results would be gratefully accepted.

We have rephrased the paragraph in between L280-295 in order to make it more readable. The text has been changed as follows: "More specifically, figure $3a_1$ compares with the reference values daytime ABLH estimates obtained through the Richardson number method (RNM) applied to the Raman lidar data, with $R^2$ being equal to 0.98 and slope being equal to 1.02. This result confirms the small bias (2 %) affecting  ABLH estimates from the Raman lidar data when compared with the reference estimates. Identical values of $R^2$ and slope are found in the night-time comparison (figure $3a_2$). Figure $3b_1$ compares with the reference values daytime ABLH estimates obtained through the RNM applied to the IGRA radiosonde data, with $R^2$ being equal to 0.94 and slope being equal to 1.01. This result confirms a very small bias (1 %) affecting ABLH estimates from the IGRA radiosonde data when compared with the reference estimates. Very similar values of $R^2$ and slope, 0.95 and 0.98, respectively, are found in the night-time

comparison (figure 3b$_2$). Figure 3c$_1$ compares with the reference values daytime ABLH estimates obtained through the RNM applied to the ECMWF-ERA5 analysis data, with R$^2$ being equal to 0.98 and slope being equal to 0.91. This result confirms a small bias (9 %) affecting the ABLH estimate from the ECMWF-ERA5 analysis data when compared with the reference estimate."

It would also be much more interesting to tentatively describe why the comparisons are better during night (caution: the relative differences are larger during night than during day), why detection based on Basil RCS and WP lead to larger (and expected) bias, what are the effect of the various vertical resolution of the profils, than to summarize the list of slopes and R2 values.

We are now providing additional information on the results, with a specific focus on the differences between day and night and with specific considerations on BASIL RCS and the Wind Profiler. The following new text has been introduced: "Results clearly reveal that absolute biases are smaller during the night than during the day. This is most probably due to the fact that night-time portions of the measurement records are typically characterized by higher stability conditions. However, percentage biases are typically larger during the night, when the ABL may become very shallow. In fact, in the presence of shallow ABLs, ABLH values become comparable with the measurement uncertainty and this makes percentage biases intrinsically high. In our specific case, ABLH values are found to not exceed 300 m throughout most part of the month of October 2012. With a specified vertical resolution of 150 m, the uncertainty affecting the estimate of the ABLH is 50 % when sounding ABLs with heights not exceeding 300 m. As already anticipated above, the bias affecting the ABLH obtained from the range-corrected elastic backscatter lidar signals is possibly generated by the limited sensitivity of this sensor below 250 m, as a result of the presence of overlap effects and a blind vertical region. Furthermore, the negative biases observed in ERA5 ABLH estimate are most probably associated with the negative bias affecting ERA5 water vapour and wind V component data."

- L301-305: long sentence.

The present long sentence has now been splitting into two shorter sentences. The text now reads: "The above results reveal that the different ABLH methods considered in the paper, which refer to different definitions and physical (dynamic and thermodynamic) processes, ultimately lead to ABLH estimates that are in very good agreement among them. This conclusion, which need to be confirmed over longer time series including a complete seasonal cycle, confirms that turbulent air motion and vertical mixing, induced by shear and buoyancy forces (Stull, 1988), cause rapid fluctuations of several physical quantities, such as flow velocity, temperature, moisture and aerosol concentration, which are strongly correlated one with the other."

Please mention that your conclusion is valid for one month of measurement at one place. It cannot be extended to an absolute statement without longer time series including a complete seasonal cycle.

The reviewer is write and this aspect is now explicitly specified in the text, where the corresponding sentence has been changed as follows: "This conclusion, which need to be confirmed over longer time series including a complete seasonal cycle, confirms that turbulent air motion and vertical mixing, induced by shear and buoyancy forces (Stull, 1988), cause rapid fluctuations of several physical quantities, such as flow velocity, temperature, moisture and aerosol concentration, which are strongly correlated one with the other."

- L316: The correlation between the friction velocity and ABLH is difficult to see from Figure 4. Why don't you directly plot one parameter as a function of the other?

Following the suggestion of the reviewer, we introduced a new plot (new figure 6) correlating ABLH with the friction velocity. This plotter was introduced the only for a BL8 sugar system friction velocity as correlations of the ABLH with the other ERA5 parameters, namely the CAPE gradient and relative humidity, had been found to be smaller. In new figure 6 the scatter plot used for the purpose of correlating ABLH with the friction velocity includes separate data points for daytime and night-time. The following new sentences have been introduced in the text: "In order to further underline the correlation between ABLH estimates and corresponding friction velocity values, a scatter plot of ABLH vs. friction velocity-ABLH) for the month of October 2012 is illustrated in figure 6. In order to reveal the higher correlation during night-time, data points are plotted separately for both daytime (red dots, 12:00 UTC) and night-time (black dots, 00:00 UTC)."

[Figure]

Figure 6: Scatter plot of ABLH vs. friction velocity for the month of October 2012. Results are reported for both daytime (red dots, 12:00 UTC) and night-time (black dots, 00:00 UTC).

- L320-322: The high stability during most of the month is probably a cause of the very good correlation between all methods and with the model. Did you compare the bias as a function of the atmospheric stability? Please comment.

Figure 5 illustrates ABLH values for the different instruments/methods/models as a function of the atmospheric stability. Indeed, a higher correlation appears to be present in case of stable weather conditions. More specifically, more than 50 % of the cases can be classified as stable condition, while the reminder cases can be classified as unstable. For the stable conditions

mutual deviations between the different/sensors methods are approximately 30% larger than in the case of unstable conditions. This aspect is now clearly specified in the text, where the following sentences have been introduced: "The above results reveal the role of atmospheric stability in limiting the variability of the ABLH. The high stability during most of the month of October is probably one of the main drivers of the very good correlation found between the different sensors/models/methods. Figure 5 illustrates the percentage bias of the different ABLH estimates with respect to the one obtained through the application of the Richardson number approach to the on-site radiosonde profiles, expressed as a function of the atmospheric stability conditions. The presence of stable or unstable conditions was verified based on the use of the "lifted index" obtained from ERA5 model analysis data, with negative values for this index indicating instability - the more negative, the more unstable the air is - and positive values indicating stability. More than 50 % of the cases can be classified as stable condition, while the reminder cases can be classified as unstable. For the unstable conditions mutual deviations between the different/sensors methods are approximately 30% larger than in the case of stable conditions. A lower dispersion of the bias values implies a higher correlation among the different ABLH estimates in case of stable weather conditions."

- L322: did you expect a correlation between the relative humidity and ABLH?

The correlation between relative humidity and ABLH was verified in order to assess the potential role of water vapor in feeding convective activity and consequently in determining an increase of the ABLH when triggering mechanisms are present. This aspect is now more clearly specified in the text, where the following sentences have been introduced: "The correlation between relative humidity and ABLH was verified in order to assess the potential role of water vapor in feeding convective activity and consequently in determining an increase of the ABLH when triggering mechanisms are present".

- L324: If I understand it correctly, the CAPE gradient = (CAPE(dayn+1)-CAPE(dayn))/24 h? Do you expect that a change in stability will correlate with the absolute value of ABLH? Why ?

We checked if changes in stability conditions were correlating with ABLH values again because, as already specified above (in the point before the previous one), we expect atmospheric stability to play a role in limiting the ABLH variability.

- L327-329: Are these different results than the ones described in the previous § ?

We are not sure we properly understand the comment/request from the review here. The results illustrated in lines 227-229 refer to the correlation between ABLH and ERA5 estimates of CAPE, friction velocity and the relative humidity. These results are reported here for the first time in the paper as previously reported results wear dealing with the correlations between the different ABLH approaches.

The previous paragraph (L316-322) was describing the time evolution of the ERA5 reanalysis data for CAPE, friction velocity and relative humidity for the entire month of October 20212 and how these parameters compare with the ABLH estimate.

Maybe the reviewer intends to refer to the fact that the correlation results are introduced here without any reference to a specific plot. In this regard, we have now corrected the corresponding sentence as follows: "We also tried to quantitative correlate ABLH estimates with the variability of CAPE, friction velocity and relative humidity. A linear fit was applied to the time series of CAPE day-by-day gradient, the friction velocity and the relative humidity values in the  time period 1-31 October 2012 vs. the corresponding ABLH estimates (plots not shown here)."

- L354: chances?

This misprint was corrected

- L 395: two "complete daily cycles" are not included.

The reviewer is right in underlying that two "complete daily cycles" are not present in the measurement record. The authors had used an erroneous wording: the expression "daily cycle" had been improperly used to mean the daytime portion of the day. We are now properly specifying in the text that we  refer to two daytime portions. The corresponding sentence has been changed as follow: " The analysis was also focused on one specific case study, covering an extended time interval from 09:00 on 18 October 2012 to 19:00 UTC on 19 October 2012, including two daytime portions, the first one characterized by clear sky conditions and the second one characterized by the present of low stratiform clouds, which allowed to assess the performance in the characterization of the short-term variability of the ABLH in variable weather conditions."

- L 395: Fig. 5a and the text do agree to describe a high altitude cloud cover during most of the 18th of October with some precipitation. How can you refer here to "clear sky conditions"

We are sorry for this erroneous wording. The corresponding sentence has been changed as follows: "The analysis was also focused on one specific case study, covering an extended time interval from 09:00 on 18 October 2012 to 19:00 UTC on 19 October 2012, including two daytime portions, the first one characterized by the presence of high scattered clouds between 3 and 4 km and the second one characterized by the present of low stratiform clouds between 1 and 2 km, which allowed to assess the performance in the characterization of the short-term variability of the ABLH in variable weather conditions."